# Reproductive barriers in cassava: Factors and implications for genetic improvement

**Massaine Bandeira e Sousa**[1☯], **Luciano Rogerio Braatz de Andrade**[1☯], **Everton Hilo de Souza**[2☯], **Alfredo Augusto Cunha Alves**[1☯], **Eder Jorge de Oliveira**[1☯]*

1 Embrapa Mandioca e Fruticultura, Cruz das Almas, BA, Brazil, 2 Center of Agrarian, Environmental and Biological Sciences, Universidade Federal do Recôncavo da Bahia, Cruz das Almas, BA, Brazil

☯ These authors contributed equally to this work.
* eder.oliveira@embrapa.br

**Data Availability Statement:** All datasets generated for this study can be found in the article, supplementary material, and Figshare (https://doi. org/10.6084/m9.figshare.14160545).

## Abstract

Cassava breeding is hampered by high flower abortion rates that prevent efficient recombination among promising clones. To better understand the factors causing flower abortion and propose strategies to overcome them, we 1) analyzed the reproductive barriers to intraspecific crossing, 2) evaluated pollen-pistil interactions to maximize hand pollination efficiency, and 3) identified the population structure of elite parental clones. From 2016 to 2018, the abortion and fertilization rates of 5,748 hand crossings involving 91 parents and 157 progenies were estimated. We used 16,300 single nucleotide polymorphism markers to study the parents' population structure via discriminant analysis of principal components, and three clusters were identified. To test for male and female effects, we used a mixed model in which the environment (month and year) was fixed, while female and male (nested to female) were random effects. Regardless of the population structure, significant parental effects were identified for abortion and fertilization rates, suggesting the existence of reproductive barriers among certain cassava clones. Matching ability between cassava parents was significant for pollen grains that adhered to the stigma surface, germinated pollen grains, and the number of fertilized ovules. Non-additive genetic effects were important to the inheritance of these traits. Pollen viability and pollen-pistil interactions in cross- and self-pollination were also investigated to characterize pollen-stigma compatibility. Various events related to pollen tube growth dynamics indicated fertilization abnormalities. These abnormalities included the reticulated deposition of callose in the pollen tube, pollen tube growth cessation in a specific region of the stylet, and low pollen grain germination rate. Generally, pollen viability and stigma receptivity varied depending on the clone and flowering stage and were lost during flowering. This study provides novel insights into cassava reproduction that can assist in practical crossing and maximize the recombination of contrasting clones.

## Introduction

Cassava (*Manihot esculenta* Crantz) is a Euphorbiaceous species that is grown in over 100 tropical and subtropical countries, where it is the fourth most important source of calories in

**Funding:** • Eder Jorge de Oliveira: CNPq (Conselho Nacional de Desenvolvimento Científico e Tecnológico). Grant number: 409229/2018-0, 442050/2019-4 and 303912/2018-9 • Eder Jorge de Oliveira: FAPESB (Fundação de Amparo à Pesquisa do Estado da Bahia). Grant number: Pronem 15/2014 • Everton Hilo de Souza: CAPES (Coordenação de Aperfeiçoamento de Pessoal de Nível Superior) Grant number: 88882.315208/ 2019-01 • Eder Jorge de Oliveira, Alfredo Augusto Cunha Alves, Massaine Bandeira e Sousa and Luciano Rogerio Braatz de Andrade: UK's Foreign, Commonwealth & Development Office (FCDO) and the Bill & Melinda Gates Foundation. Grant number: INV-007637 • The funder provided support in the form of fellowship and funds for the research, but did not have any additional role in the study design, data collection and analysis, decision to publish, or preparation of the manuscript.

**Competing interests:** The authors declare that the research was conducted in the absence of any commercial or financial relationships that could be construed as a potential conflict of interest.

the diets of approximately 800 million people [1]. It is primarily cultivated by small-scale farmers in marginal areas with low-fertility soils and long drought periods. The high root yield of cassava in these regions provides food and some income for farmers living in areas prone to climatic challenges.

Since cassava is most often clonally propagated, there is no genetic variation over the generations. However, cassava can also be propagated using botanical seeds (obtained by sexual reproduction), resulting in new genotypic combinations that may improve agronomic performance in comparison with the mother plant. Many landraces grown throughout Brazil were generated from seedling selection in production areas. Thus, breeders can use sexual reproduction to produce segregating populations of new genotypes for breeding. In conventional breeding, seeds are produced through self-pollination or controlled crossings and then used as a source of genetic variation for the selection of superior genotypes and development of partially inbred lines [2–4].

A major challenge for planning crosses is the lack of flowering synchronization, since some clones bloom early (4–5 months after planting, MAP) and others bloom later (>10 MAP) [5]. Therefore, to plant crossing block fields and guarantee good flowering synchronization, it is important to understand the flowering phenology of the parents. A recent study involving over 1,000 cassava accessions showed high phenological diversity of flowering and fruiting performance in Embrapa's cassava germplasm [6].

Flowering is a complex process affected by environmental (i.e., temperature, humidity, photoperiod, and altitude) and genetic factors [7]. During the transition from the vegetative to reproductive stage, endogenous signaling is required [8]. Several techniques have been applied to induce flowering in cassava, including the exogenous application of growth regulators (e.g., silver thiosulfate, STS) [9], grafting [5, 10], and the development of transgenic plants that overexpress the flowering locus T (FT) gene [11]. The *MeFT1* and *MeFT2* genes (members of the phosphatidylethanolamine-binding protein (PEBP) family to which the FT gene also belongs) have been associated with photoperiod signaling and early cassava flowering in gene expression studies [12].

Despite reports of successful flowering induction in cassava, occurrences of high rates of flower abortion (84.4 to 95.0%) during the generation of cassava progenies demand huge labor to achieve a satisfactory population size for an efficient recombination [13, 14]. Reproductive studies may clarify whether the high rate of flower abortion in some cases is associated with the presence of pre-zygotic and/or post zygotic reproductive barriers [15], and/or environmental factors [16]. Although reproductive barriers are commonly observed in interspecific crosses, there are reports of male sterility in various intraspecific crosses [17]. Some pre- and post-zygotic barriers include: abnormalities in the basal region of the style; different style and pollen tube lengths; inadequate pollen grain morphology and size; no pollen germination in the stigma due to insufficient hydration or nutrients; early embryo and endosperm degeneration; and hybrid sterility [18].

In addition to the high rates of flower abortion, the generation of new segregating cassava populations has been limited due to poor flowering ability of certain genotypes [5, 14]. To date, there is no evidence of incompatibility between genotypes or self-incompatibility in cassava [13]. However, wide variation in hybrid fertility between families has been observed [19].

Previous studies have reported pollen grain trimorphism, a negative correlation between pollen grain viability and size, and the gradual death of pollen grains with advanced maturity —all of which result in reduced pollen viability at anthesis [20]. Other factors also influence the efficiency of sexual reproduction in cassava, such as pollen-pistil interaction and the influence of environmental factors during pollination. However, these factors have not yet been clarified due to the low number of crosses evaluated. Therefore, identifying barriers to

fertilization and how to overcome them is crucial to selecting the best time for pollination and maximizing the generation of new clones. This study aimed to: 1) evaluate the influence of abiotic factors on the abortion rate of cassava flowers; 2) analyze the presence of reproductive barriers in controlled crosses; and 3) evaluate pollen-pistil interaction at three stages of floral development versus the population structure of different cassava genotypes.

## Materials and methods

### Plant material and experimental procedures

A total of 91 cassava genotypes were used as parents for controlled crosses: 10 improved cultivars, 15 improved lines obtained by Embrapa's breeding program, and 66 landraces from Embrapa's gene bank. The origin and agronomic attributes of these genotypes are presented in S1 Table. The parents varied in terms of root and starch production, dry matter content, and disease resistance traits. For this reason, they have been routinely used by Embrapa's cassava breeding program for the development of segregating populations.

The genotypes were grown in two crossing block fields located in the experimental area of Embrapa Mandioca e Fruticultura in Cruz das Almas, Bahia, Brazil (12˚39′25″ S, 39˚07′27″W, 226 m altitude). The first field was planted from 2015 to 2017 and the second from 2016 to 2018. The region's weather conditions are tropical hot and humid (Aw/Am, according to the Köppen classification), with a photoperiod throughout the year of approximately 12 hours [6]. Cuttings (16–20 cm long) with 5–7 buds were grown under rainfed conditions in plots containing two rows with eight plants each, spaced 1.20 m between rows and 0.80 m between plants. All cultivation practices were adopted from [21]. Basic climatic data were collected daily at the weather station of Embrapa Mandioca e Fruticultura throughout the experimental period.

### Evaluation of the abortion and seed setting rates in controlled crosses (Experiment 1)

The controlled crosses were randomly performed among 91 cassava genotypes to obtain $F_1$ seeds (Table 1 and S1 Table). Overall, 43 were used as both male and female parents, while 42 and 6 were used only as female or male parents, respectively. All told, 25,521 female flowers (pistillate) from 7,608 inflorescences were pollinated. From 2016 to 2018 we performed around 2,751 crosses per year. To prevent insect pollination, the female flowers were protected by a voile fabric bag 24 hours before anthesis, which can be easily identified by field workers experienced in cassava crossing.

Male flowers were collected during the morning of the anthesis day (7–9 am), while the crosses were performed between 9 am and 4 pm by distributing the fresh pollen grains on the

**Table 1. Number of genotypes used as parents in controlled crosses and the number of flowers, hybrid combinations, and characteristics assessed in each experiment.**

| Experiment | Number of female parents | Number of male parents | Number of flowers | Number of hybrid combinations | Traits assessed |
|---|---|---|---|---|---|
| 1 | 85 | 49 | 25,521 | 157 | • Number of seeds<br>• Abortion rate |
| 2 | 21 | 19 | 1,037 | 130 | • Pollen grains adhered to the stigma surface<br>• Germinated pollen grains<br>• Pollen tube growth<br>• Number of ovules fertilized |

stigmas. One male flower was used to pollinate up to three female flowers, depending on the amount of pollen available. The female flowers were protected again shortly after being pollinated, as described above. A cross was defined as a single pollination event. After identifying the female flowers ready for pollination, the crosses were performed in one to four flowers per inflorescence, and the remaining flowers were removed. The protective bag covered the inflorescence until the seeds were released and collected, which occurred approximately 2 to 3 months after pollination.

The traits evaluated in Experiment 1 were analyzed each 10 days to capture the environmental effects on flower abortion, such as differences in pluviosity, temperature and relative humidity. The numbers of pollinated flowers and seeds obtained were noted for each controlled pollination. Therefore, two main parameters were assessed in Experiment 1 (2015–2018). The first was the seed setting rate $\left(SS_w = 100 \times \frac{SS}{N.cross}\right)$, in which $SS$ corresponds to the total seed setting produced over a period of 10 days per progeny; and $N. cross$ is the number of crossings performed. The $SS_w$ from different times was recorded with the objective of evaluating how the environment at that time could affect the seed setting. In summary, for each 10 days, the $SS_w$ was recorded for to study the correlation with the weather variables. The second was the abortion rate (%), estimated by the formula $AR_w = 1 - \left[100 \times \left(\frac{NS}{3 \times NF}\right)\right]$, where $NS$ is the number of seeds produced over 10 days; and $NF$ is the number of flowers pollinated during 10 days. The number three in the $AR_w$ formula is explained by the fact that under natural conditions, the cassava pistil is trilocular, with a single ovule in each locule, producing a maximum of three seeds per flower.

## Statistical analysis of Experiment 1

Due to unbalanced crosses in Experiment 1, Genetic Design 1 from Comstock and Robinson [22] was used to evaluate the male and female effects of the controlled crosses performed in Experiment 1, according to the model: $y_{ijkl} = Dec_k + Dec \times Y_{kl} + F_i + M/F_{i(j)} + \varepsilon_{ijkl}$, where $y_{ijkl}$ are the phenotypic observations ($SS_w$ and $AR_w$) of the $i^{th}$ female and $j^{th}$ male in the $k^{th}$ 10-day intervals and $l^{th}$ year; $Dec_k$ is the fixed effect of the 10-day intervals; $\hat{Dec} \times \hat{Y}_{kl}$ is the effect of the interaction of the 10-day intervals and year (random effect); $M_i$ and $F_j$ are the random effects of male and female, respectively (assumed to be $M \sim N(0, \sigma_M^2)$ and $F \sim N(0, \sigma_F^2)$, respectively), where $\sigma_M^2$ and $\sigma_F^2$ are the variances of male and female effects, respectively; $M/F_{i(j)}$ corresponds to the effect of the $j^{th}$ male within the $i^{th}$ female; and $\varepsilon_{ijkl}$ is the random effect of the experimental error, assumed as $\varepsilon \sim N(0, \sigma_\varepsilon^2)$, where $\sigma_\varepsilon^2$ is the error variance.

Possible relationships of weather variables with abortion rate ($AR_w$) and seed setting rate ($SS_w$) were investigated using cubic regression based on average climatic data: temperature (minimum, average, and maximum); relative humidity (%); and accumulated rainfall for each 10-day intervals. The weather parameters were considered a fixed effect in the model: $y_{ijk} = \beta_0 + \beta_1 X_k + \beta_2 X_k^2 + \beta_3 X_k^3 + F_i + M/F_{i(j)} + \varepsilon_{ijk}$, where $y_{ijk}$ is the phenotypic observation of the $i^{th}$ female and $j^{th}$ male in the $k^{th}$ 10-day intervals; $X$ is the independent variable of minimum, average, and maximum temperature (˚C), relative humidity (%), and rainfall (mm) for each 10-day intervals; $M_i$ and $F_j$ are the random effects of male and female, respectively (assumed to be $M \sim N(0, \sigma_M^2)$ and $F \sim N(0, \sigma_F^2)$, respectively), where $\sigma_M^2$ and $\sigma_F^2$ are the variance of males and females, respectively; $M/F_{i(j)}$ corresponds to the effect of the $i^{th}$ female within the $j^{th}$ male parent; $\varepsilon_{ijk}$ is the random effect of the experimental error, assumed to be $\varepsilon \sim N(0, \sigma_\varepsilon^2)$, where $\sigma_\varepsilon^2$ is the error variance; and $\beta_0$, $\beta_1$, $\beta_2$, and $\beta_3$ are the constant, linear, quadratic, and cubic regression coefficients for weather variables. Statistical analyses were performed using the lme4 package [23] of R version 4.0.2 [24].

## Pollen tube development in the pistil, stigma receptivity and pollen viability assessment via histochemical analysis (Experiment 2)

To obtain a better understanding of the pre-zygotic reproductive barriers that affect the abortion and seed setting rates in cassava, 39 parents were selected from the results of Experiment 1 to obtain a more detailed analysis of pollen tube development and to estimate the maternal and paternal effects. However, after a thorough analysis in the field, only 21 of the 39 parents flowered during the pollinations (Table 1). The following criteria were considered when choosing parents for Experiment 2: 1) maternal genotypes with low, medium, and high reproductive success; and 2) paternal genotypes with high, low, or no reproductive success, both based on the values of $SS_w$ and $AR_w$. After selecting these genotypes, another 135 random cross combinations were performed using 1,037 female flowers and fresh pollen grain.

Stigma receptivity was analyzed *in vivo* considering three separate anthesis periods: pre-anthesis; anthesis; and post-anthesis. Three to five replicates (i.e., flowers) were evaluated for each parent combination and anthesis period. Pre-anthesis was defined as almost mature flowers (one day before anthesis), anthesis as open and functional flowers (up to 4 hours after opening), and post-anthesis as fully open flowers evaluated one day after anthesis.

The flowers were identified, their anthers were removed, and the female flowers were pollinated using fresh pollen grain and protected as described in Experiment 1. Then, 48 hours after pollination, the pistils were collected and fixed in Carnoy's solution (3 parts ethanol (95%): 1 part acid glacial acetic) for 48 hours, followed by clearing in sodium sulfite solution (10%) and autoclaved for 20 min at 120°C, in order to soften the tissues. The material was stained overnight with aniline blue solution (0.01%) in tribasic phosphate buffer. To verify the germination of pollen grains in the stigma and the development of pollen tubes along the pistil, fluorescence microscopy with ultraviolet filtering was used [18, 25, 26]. The slides were analyzed and photographed under a BX51 fluorescence microscope (Olympus Latin America Inc.).

To evaluate the germination of fresh pollen grains *in vivo* and the growth of pollen tubes in the pistils of cassava crosses, four traits were analyzed: i) the number of pollen grains that adhered to the surface of the stigma (PGA); ii) the number of pollen grains that germinated on the surface of the stigma (PGG); iii) pollen tube growth in the pistil (PTG); and iv) the number of fertilized ovules (NFO). PGA was evaluated according to the following scale of pollen grain presence: 1) 1 to 5 pollen grains; 2) 6 to 25 pollen grains; 3) 26 or more pollen grains. PGG was measured based on the following scale of pollen grain germination: 0) no germinated pollen grains; 1) 1 to 5 pollen grains germinated; 2) 6 to 25 pollen grains germinated 3) 26 or more pollen grains germinated. To assess pollen tube growth in the pistil, specific regions were considered according to [13] with some minor adaptations. Pollen tube growth was measured based on the following scale: 0) no pollen grains germinated on the surface of the stigma; 1) some pollen grains germinated on the surface of the stigma; 2) pollen tubes grew in the stylet; 3) pollen tubes grew in the ovary; 4) pollen tubes grew close to the ovary; and 5) pollen tubes penetrated the micropyle.

Pollen grain viability was evaluated in 16 of the 21 male parents in Experiment 2 to identify a possible correlation with the development of the pollen tube in the pistil. For histochemical evaluation, fresh pollen grains were collected 3–4 hours before anthesis from three flowers of each genotype. The grains were then distributed on glass slides and a drop of Alexander's 2% lactic acid solution was added [27]. The percentage of viable pollen grains was determined by counting the dark purple grains (viable ones) in relation to the green (non-viable) grains in three replicates, with each replicate being represented by a slide prepared using one of three flowers. Slides were analyzed and photographed in a bright field under a fluorescence microscope (Olympus Latin America Inc.).

## Statistical analysis of Experiment 2

The following mixed linear model was considered for Experiment 2: $y_{ijklm} = AP_k + GPA_l + p_{ij} + f_i \times m_j + \varepsilon_{ijklm}$, where $y_{ijklm}$ is the phenotypic observation of the $k^{th}$ anthesis period from the $i^{th}$ female flower with the $l^{th}$ class of pollen grain amount adhered to the stigma for the $m^{th}$ replicate; $AP_k$ is the fixed effect of anthesis period; $GPA_l$ is the fixed effect of the number of pollen grains that adhered to the stigma; $p_{ij}$ is the $i^{th}$ female and $j^{th}$ male parent effect, respectively, with $p_{ij} = f_i + m_j$, $f_i$ and $m_j$ is the $i^{th}$ female effect and $j^{th}$ male effect, respectively; $p_{ij}$ was assumed to be a random effect with the following parameters: $p_{ij} \sim N(0, \sigma_p^2)$, where $\sigma_p^2$ is the variance of the parent effect; $f_i \times m_j$ is the female-male interaction (F×M) of the $i^{th}$ female and $j^{th}$ male, assumed to be a random effect with the parameters: $f_i \times m_j \sim N(0, \sigma_{fxm}^2)$, where $\sigma_{fxm}^2$ is the variance of the F×M interaction effect; $\varepsilon_{ijklm}$ is the experimental error effect (assumed to be random) with the following parameters: $\varepsilon \sim N(0, \sigma_\varepsilon^2)$, where $\sigma_\varepsilon^2$ is the residual variance. The mixed linear model was run using the sommer package [28] for R version 4.0.2 [24].

The self-incompatibility index (SII) was estimated based on parents with self-fertilization in Experiment 2 only. SII is defined by the equation $SII = SS_i / SS_o$, where $SS_i$ is the relative success of self-pollination, defined as the ratio between the set of flowers with the condition of at least one fertilized ovule and the total number of self-fertilized flowers, while $SS_o$ is the relative success of cross-pollination, defined as the ratio between the set of flowers containing at least one fertilized ovule and the total number of crossed flowers. For statistical analysis, the genotypes were grouped into three categories based on the SII index: 1) self-incompatible, with SII < 0.30; 2) partially self-incompatible, with SII between 0.30 and 1.00; and 3) self-compatible, with SII > 1.00 [29].

## Genomic data, population structure analysis and putative heterotic groups

The DNA of cassava parents was extracted using the cetyltrimethylammonium bromide (CTAB) protocol described by [30]. DNA samples were sent to the Genomic Diversity Facility at Cornell University (http://www.biotech.cornell.edu/brc/genomic-diversity-facility) for genotyping by sequencing (GBS). Individual DNA samples were digested with the restriction enzyme *Ape*KI, linked to a barcode, and multiplexed into 95-plex libraries [31]. Libraries were sequenced on the Illumina HiSeq2500 platform. Subsequent sequence reads were demultiplexed and aligned with the cassava reference genome v.6 [32] using BWA alignment [33]. Single-nucleotide polymorphism (SNP) calling was then performed using the TASSEL GBS pipeline [34]. The threshold for minimum SNP read depth was five, during the TASSEL pipeline analysis. A total of 27,045 SNPs distributed across all 18 cassava chromosomes were obtained. Genomic data were subjected to quality control via call rates ≤ 0.80 and lower minor allele frequency (MAF) < 0.05. After quality control, the matrix of markers comprised 16,300 SNPs for the parents evaluated in Experiment 1.

SNP markers were used to estimate the genomic kinship matrix among the parents of Experiment 1 (n = 86) using van Raden's method [35]. A smaller number of individuals were used in the molecular analysis because not all genotypes used in Experiment 1 had molecular data.

Parental population structure was determined by discriminant analysis of principal components (DAPC) using the adegenet package [36] for R version 4.0.2 [24]. The number of clusters in the population was detected using the *find.clusters* function. This function uses K-means grouping, which decomposes the total variation of each variable into components between and within groups. The best number of subpopulations was the one with the lowest Bayesian information criterion. The groups were then plotted on a scatterplot of the first and second linear discriminants of the DAPC.

Cassava parents were clustered based on principal component analysis (PCA) as well as considering the effects of the male and the F×M effects obtained from Experiments 1 and 2. These groups were used to identify putative heterotic groups to improve seed production, optimize breeding program resources, and generate larger segregating populations in a short time.

## Results

### Naturally high abortion rate and low seed setting rate in cassava (Experiment 1)

The first step of this work was to investigate the parameters $SS_w$ and $AR_w$ in the crossings carried out from 2016 to 2018. A total of 157 $F_1$ progenies were obtained from the planned crosses among 91 cassava parents (Table 1). Of the 25,521 fertilized flowers, 76,576 seeds (trilocular fruits) were expected, but only 11,445 seeds were collected (i.e., 38.14% of the maximum number of seeds). The average abortion rate per progeny was 88.0%, ranging from 46.8% to 100.0%. Most of the progenies (94.6%) presented flower abortion rates higher than 75% (S1 and S2 Figs).

Six progenies from crosses between 10 different parents had a 100% abortion rate. The crosses between parental genotypes BGM-0968 and BGM-1819 yielded the largest seed setting (643 seeds from 41% of successful crossings). The effects of female and male within female parents were significant for abortion rate and seed setting rate, which suggests the presence of genetic reproductive barriers among the analyzed parents (S2 Table). Therefore, additional pollen viability and stigma receptivity tests performed according to Experiment 2 are essential to prove the fertilization results from these historical crossing datasets.

### Flower abortion and seed setting rates of cassava crosses are influenced by genetic effects and high temperatures (Experiment 1)

Due to the wide variation of the parameters $SS_w$ and $AR_w$ and the fact the field data were evaluated in different periods of the year (10-day intervals), the second part of the work was to investigate the possible relationships between weather variables and the variation of $SS_w$ and $AR_w$. In the crossings from 2016 to 2018, the seed setting rate was significantly affected by rainfall and temperature (minimum, average, and maximum), while the rate of flower abortion was more strongly influenced by the maximum temperature (Table 2). Notably, the female flower abortion rate increased at temperatures above 30˚C.

The cubic regressions demonstrated that an increase in the minimum, average, and maximum temperatures resulted in a major reduction in seed setting (up to -92.63%). For example, increasing the maximum temperature from 24 to 34˚C reduced the seed setting per 100 crosses from 190 to 14 (Fig 1). On the other hand, the effect of rainfall per 10-day intervals on seed setting was more stable between 0 to 80 mm of rainfall, with a significant increase in the seed setting with rainfall greater than 80 mm per 10-day intervals. Relative humidity had no association with the abortion or seed setting rate.

In the Cruz das Almas region (Bahia, Brazil), the highest seed setting occurred during the rainy season, which coincides with milder temperatures during winter (Fig 2). A severe reduction in the seed setting rate occurred as temperatures increased (after September) and rainfall decreased, which was possibly due to the high abortion rates observed among the crossings performed during spring-summer in the Cruz das Almas region. With the reduction of environmental stress effects, the abortion rate of flowers and fruits also declined, which contributed to the increased seed setting.

**Table 2. Cubic regression analysis for seed setting and abortion rates with environmental variables in controlled crossings involving 91 cassava parents.**

| Random Effects | DF | LRT—Abortion rate (%) | | | | |
|---|---|---|---|---|---|---|
| | | Rainfall | minT | averageT | maxT | RH |
| Male + Female x Male | 1 | 16.61* | 12.56* | 11.88* | 12.29* | 5.58* |
| Female | 1 | 11.98* | 13.02* | 11.53* | 10.73* | 14.42* |
| | | Mean Squares | | | | |
| Regression | 3 | 317.84 | 249.17 | 423.02 | 592.03* | 381.03 |
| Residual | 488 | 177.55 | 179.42 | 179.37 | 178.46 | 184.63 |
| CV (%) | | 15.72 | 15.81 | 15.80 | 15.76 | 16.03 |
| Random Effects | DF | LRT—Seed set rate | | | | |
| | | Rainfall | minT | averageT | maxT | RH |
| Male + Female x Male | 1 | 20.49* | 11.48* | 10.57* | 11.06* | 5.84* |
| Female | 1 | 7.64* | 5.71* | 5.11* | 5.01* | 9.31* |
| Anova | | Mean Squares | | | | |
| Regression | 3 | 65,448.00* | 129,643.00* | 170,721.00* | 165,437.00* | 15,027.00 |
| Residual | 488 | 18,898.25 | 19,162.77 | 19,039.92 | 19,070.66 | 23,463.80 |
| CV (%) | | 90.97 | 91.60 | 91.31 | 91.38 | 101.36 |

* p<0.05. minT: minimum temperature; averageT: average temperature; maxT: maximum temperature; RH: relative humidity.

## Pollen grain viability is genotype-dependent

Among the 21 parents evaluated in Experiment 2, five did not present male flowers during the pollen viability tests. According to the histochemical analysis of pollen grains, there was a significant difference (p<0.001) between the cassava genotypes analyzed for pollen viability (Fig 3), whose percentage varied from 13.33% (clone 7909–4—Fig 3C) to 96.66% (Aipim Abacate—Fig 3B). Only four genotypes exhibited pollen viability greater than 80% (Aipim Abacate, BRS Novo Horizonte, BGM 2120, and BGM-0685). Furthermore, trimorphism was observed in the BRS Novo Horizonte genotype, which produced small, medium, and large pollen grains (Fig 3D). Only violet-colored pollen grains were viable, showing the growth capacity of the pollen tube in the pistil. In general, no association was found between pollen viability and level of improvement (i.e., improved lines × landraces) due to the large variability of improved varieties and landraces in terms of pollen viability (Fig 3A).

Estimates of pollen viability did not explain the high abortion rates of crosses due to the lack of a relationship between pollen viability and pollen grain germination, since some clones with high pollen viability (>50%) had no pollen grain germination, including Aipim Abacate, BRS Novo Horizonte, BGM-2338, BGM-0661, and BGM-0685 (Table 3 and S3 Table). The exceptions were 7909–4 and BRS Tapioqueira clones, which showed high pollen viability estimates and number of pollen grains germinated in Experiment 2. Therefore, pollen viability is not the only factor that explains the high abortion rate of female flowers in cassava.

## Low pollen tube growth in cassava affects crossing viability (Experiment 2)

The results of Experiment 1 improved the understanding about the environmental effect and the important genetic variation in the seed setting and abortion rates of cassava. However, more detailed studies of pollen quality and pollen tube growth for fertilization and seed production were conducted to better understand the pre-zygotic reproductive barriers that affect the abortion and seed setting rates in cassava.

Based on the ranking of 91 cassava parents (Experiment 1 –S4 and S5 Tables), 39 contrasting parents for the abortion rate and seed setting traits were selected for further microscopic

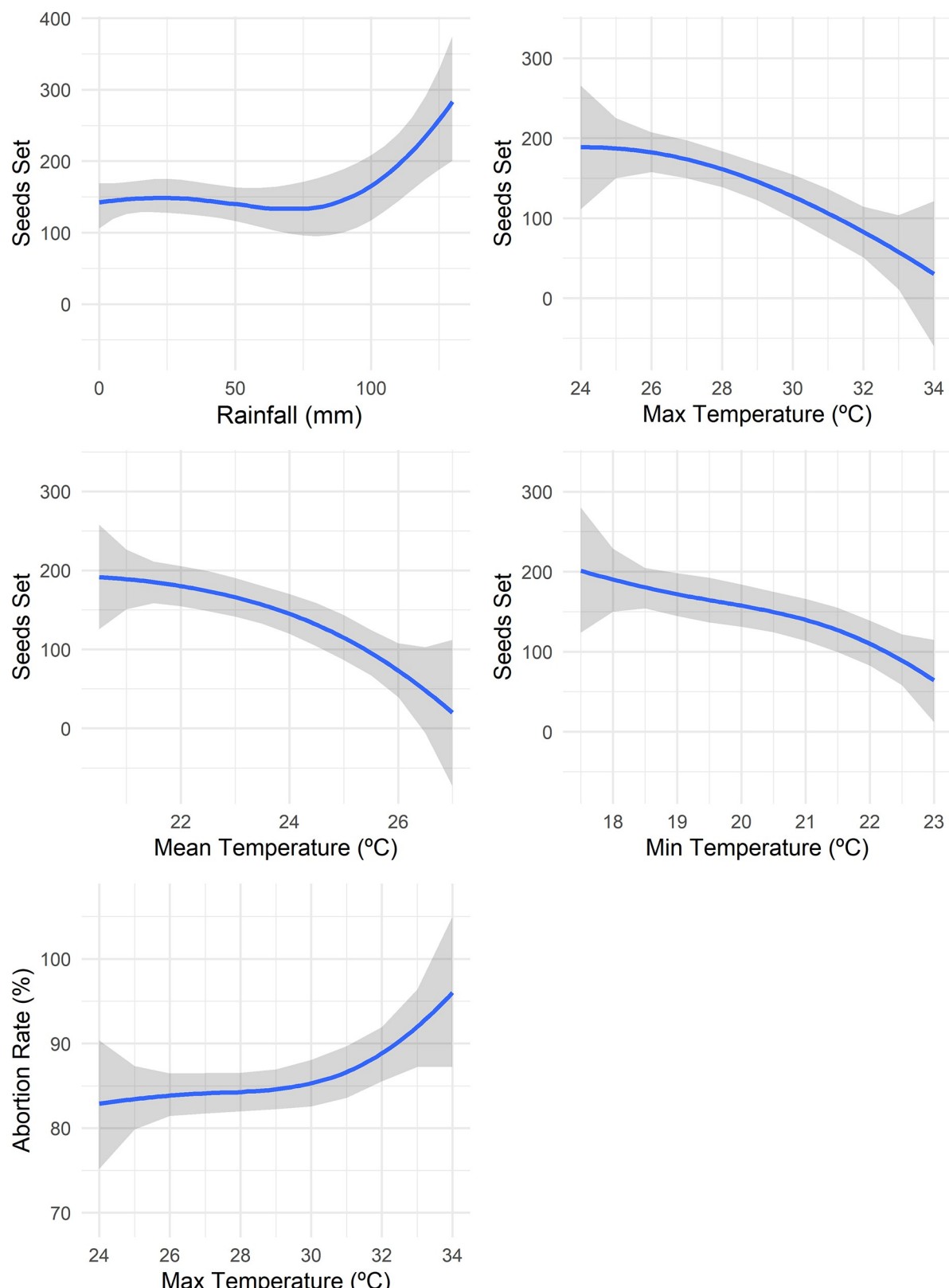

**Fig 1. Cubic regression of weather variables (rainfall and minimum, average, and maximum temperatures) for the seed setting rate (seeds produced every hundred crossings) and maximum temperature for abortion rate.** The blue line is the cubic regression line, while the gray area refers to the confidence interval of the regression.

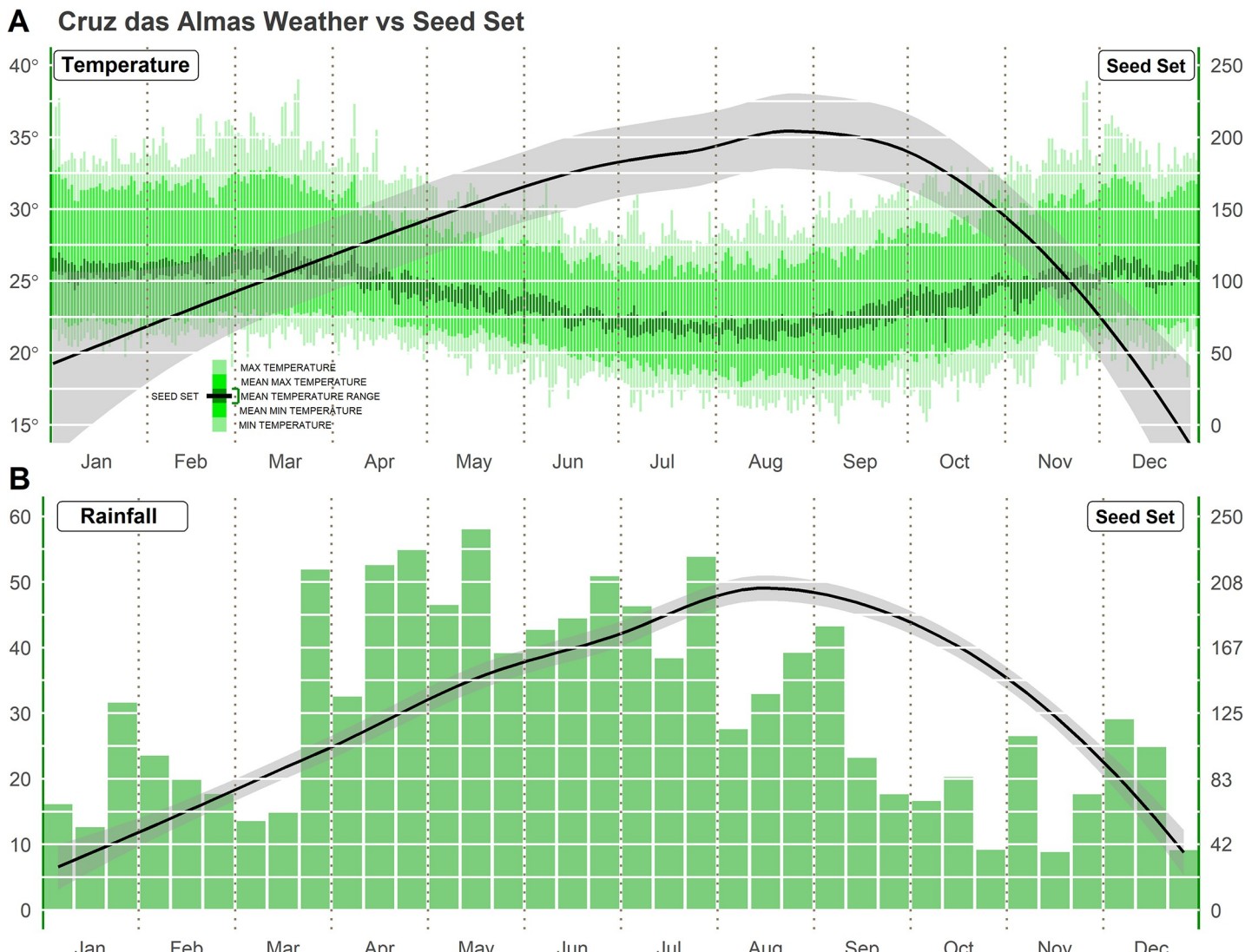

**Fig 2. Average weather data for Cruz das Almas, Bahia, Brazil between 2010 and 2019 and the LOESS regression for average rainfall and temperature for seed setting.** (A) Effect of average temperature on seed setting: light green–maximum and minimum daily temperatures; green–average maximum and minimum daily temperatures; dark green–confidence interval for average daily temperature. (B) Effect of accumulated rainfall per 10-day intervals (mm) on seed setting.

analysis to ascertain the pollen viability of male parents and its interaction in the ovule fertilization of female parents. After selecting these genotypes, another 135 random cross combinations among 21 of the 39 selected parents were performed using 1,037 female flowers. Development of the pollen tube in the pistil was observed in controlled crosses as well as in self-fertilizations. Irregular callose plates and fluorescent rings in the pollen tubes were observed during the analysis of pollen tube growth in different regions of the pistil (Fig 4).

The irregular deposition of callose plates in the cassava genotypes indicates partial incompatibility, since the sequential deposition of callose generally provides greater pollination success. Pollen grain dimorphism was observed in 90% of the genotypes (Fig 4A) and variation in pollen grain size was directly related to their germination ability. Larger pollen grains showed greater viability and germination ability, while smaller pollen grains were inviable, resulting in the absence of emission and pollen tube growth (Fig 4B–4D). Pollenkitt on the surface of the

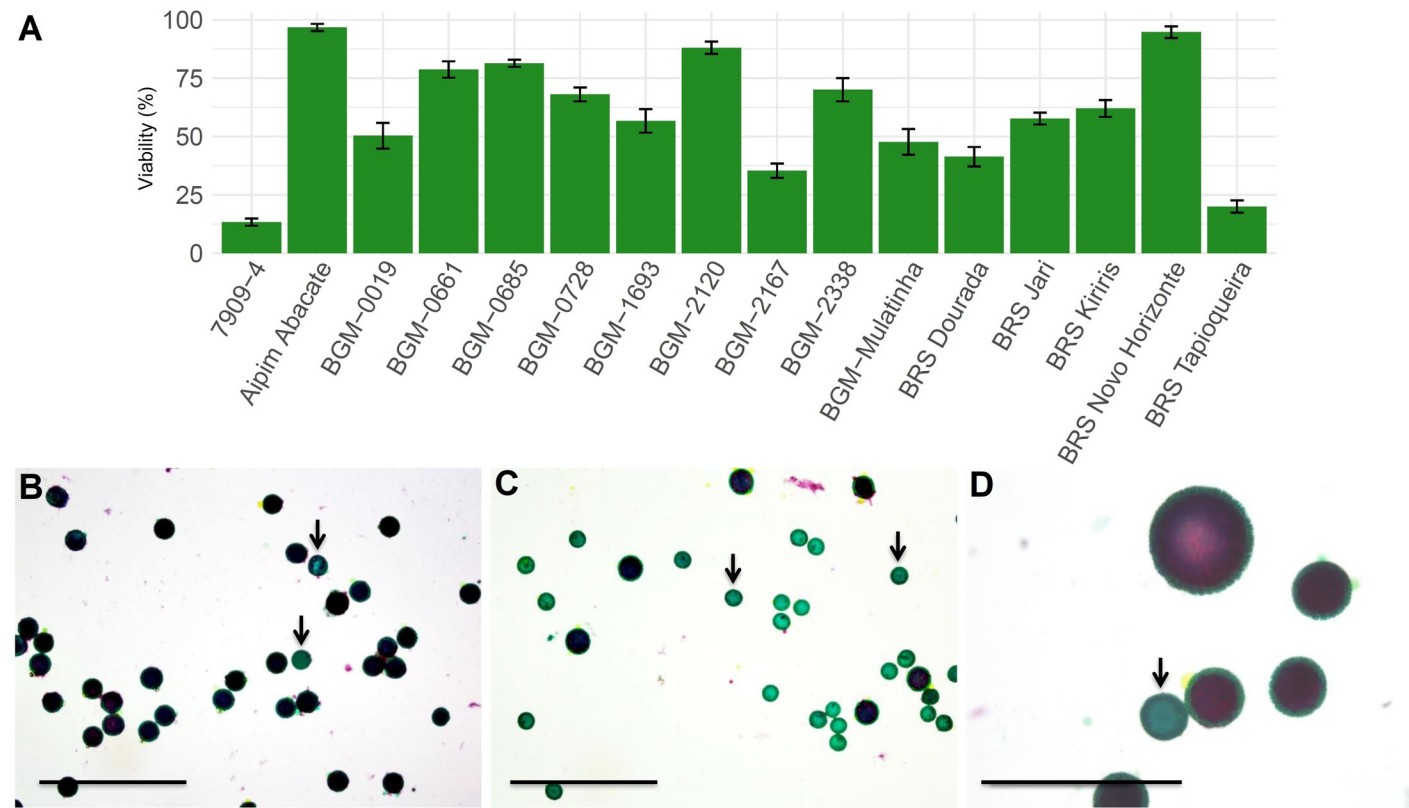

**Fig 3. Pollen viability of 16 cassava genotypes evaluated with 2% Alexander's solution.** (A) Average and standard deviation of pollen viability of the different genotypes; (B) high pollen viability in the Aipim Abacate genotype; (C) low pollen viability in clone 7909–04, with viable (violet) and non-viable (green—arrow) pollen grains; (D) trimorphism of pollen grains in the BRS Novo Horizonte genotype. Bars: 1 mm.

pollen grains was confirmed in all analyzed genotypes (Fig 4E and 4F). Pollenkitt is generally useful for adherence of pollen grains to the stigma.

Among the 1,056 evaluated crosses, 35.7% presented pollen grains that germinated on the surface of the stigmatic papillae. In 5.9% of these crosses, pollen grains germinated without further growth of the tube in the stylar region, which suggests some level of incompatibility (Fig 4C–4F). The cessation of pollen tube growth in the stylet was observed in only 0.38% of crosses (Fig 4G), and no pollen tubes were observed in the ovary region or penetrating the micropyle (Fig 4K and 4L).

Of the 376 crosses with germinated pollen grains, 315 involved the fertilization of at least one ovule (Fig 4K–4N and S6 Table), even with irregular callose deposition (Fig 4H–4J). In most cases, there was a positive relationship between the number of pollen grains that adhered to the stigma and the number of germinated pollen grains (S6 Table). The relationship between the number of germinated pollen grains and the pollen tube growth was high and suggests the presence of the SII sporophytic type, due to the high rate of non-germinated pollen grains observed in Experiment 2 (64.40%), and almost all germinated pollen grains penetrated the micropyle (83.77% of the germinated grains).

A detailed analysis of *in vivo* pollen grain germination and the pollen tube growth in the pistils of cassava crosses identified some incompatibility barriers. The analysis of variance and likelihood ratio test of the dataset generated in Experiment 2 revealed significant differences of the traits number of pollen grains that adhered to the stigma surface, number of germinated

**Table 3. Observed frequencies of pollen grains adhered to the stigma and germinated in different cassava genotypes used as male parents in Experiment 2.**

| Clone | Pollen grains adhered to stigma (%) | | | Pollen grains germinated (%) | | | |
|---|---|---|---|---|---|---|---|
| | 1 | 2 | 3 | 0 | 1 | 2 | 3 |
| 7909–4 | 80.0 | 18.7 | 1.3 | 69.3 | 21.3 | 9.3 | - |
| Aipim Abacate | 62.5 | 20.0 | 17.5 | 60.0 | 35.0 | 5.0 | - |
| BGM-0019 | 20.9 | 22.4 | 56.7 | 25.4 | 37.3 | 23.9 | 13.4 |
| BGM-0470 | - | - | 100.0 | 45.5 | 45.5 | 9.1 | - |
| BGM-0661 | 16.1 | 24.1 | 59.9 | 59.1 | 19.7 | 17.5 | 3.6 |
| BGM-0685 | 9.1 | - | 90.9 | 50.0 | 18.2 | - | 31.8 |
| BGM-0728 | 23.0 | 24.3 | 52.7 | 41.9 | 29.7 | 2.7 | 25.7 |
| BGM-1693 | 25.0 | 35.0 | 40.0 | 90.0 | 10.0 | - | - |
| BGM-1760 | 65.4 | 27.2 | 7.4 | 74.1 | 22.2 | 3.7 | - |
| BGM-2020 | 71.4 | 14.3 | 14.3 | 85.7 | 14.3 | - | - |
| BGM-2120 | 33.3 | 66.7 | - | 100.0 | - | - | - |
| BGM-2167 | 27.3 | 54.5 | 18.2 | 81.8 | 18.2 | - | - |
| BGM-2338 | 80.0 | 13.8 | 6.2 | 50.8 | 44.6 | 4.6 | - |
| BRS Dourada | 54.9 | 30.8 | 14.3 | 79.1 | 20.9 | - | - |
| BRS Jari | 50.7 | 38.4 | 11.0 | 84.9 | 13.7 | 1.4 | - |
| BRS Kiriris | 60.7 | 29.8 | 9.5 | 82.1 | 17.9 | - | - |
| BRS Novo Horizonte | 64.0 | 24.7 | 11.3 | 65.3 | 28.0 | 6.7 | - |
| BRS Tapioqueira | 61.5 | 30.8 | 7.7 | 50.0 | 38.5 | 11.5 | - |
| BRS Verdinha | 71.4 | 28.6 | - | 81.0 | 19.0 | - | - |

Pollen grains adhered to stigma: 1) 1 to 5 pollen grains; 2) 6 to 25 pollen grains; 3) 26 or more pollen grains.

Pollen grains germinated: 0) no germinated pollen grains; 1) 1 to 5 germinated pollen grains; 2) 6 to 25 germinated pollen grains; 3) 26 or more germinated pollen grains.

pollen grains, pollen tube growth, and number of fertilized ovules per flower in the different crosses among the 21 selected parents (Table 4 and S7 Table).

Basic information on the reproductive biology of cassava—especially the period of stigma receptivity—is important to support controlled crosses for breeding purposes and molecular studies. The crosses performed at post-anthesis (20–24 hours after anthesis) showed a slight increase in pollen grain germination on the stigma surface compared to pre-anthesis and anthesis (S8 Table). The effect of the number of pollen grains adhered to the stigma surface analyzed at anthesis exhibited high significance for all traits related to reproductive capacity, e.g., PGG, PTG, and NFO (Table 4). The number of pollen grains adhered to the stigma surface (PGA) is an important factor affecting reproductive capacity, since the greater number of adhered grains results in increased fertilization, as measured by the traits PGG, PTG and NFO (S8 Table). For example, in cases where higher PGA values were involved in fertilization, there was also a higher PGG (0.86), PTG (2.29) and NFO (1.11) compared to the lower PGA, which resulted in lower values of PGG (0.30), PTG (1.23) and NFO (0.48) (S8 Table). Crossings performed during the post-anthesis period showed a slight increase in pollen tube growth, in the number of pollen grains that germinated on the stigma, and in the NFO per female flower (S8 Table). Additionally, the female flowers pollinated in the anthesis period resulted in fewer germinated pollen grains, early stopping of pollen tube growth, and a lower number of embryo sacs fertilized in female flowers (S8 Table).

## Non-additive effects controlling crossability in cassava

The cassava parent effects were significant for PGA, PGG, and NFO, with greatest importance for PGA due to the high proportion between the parents and F×M variances. The effect of

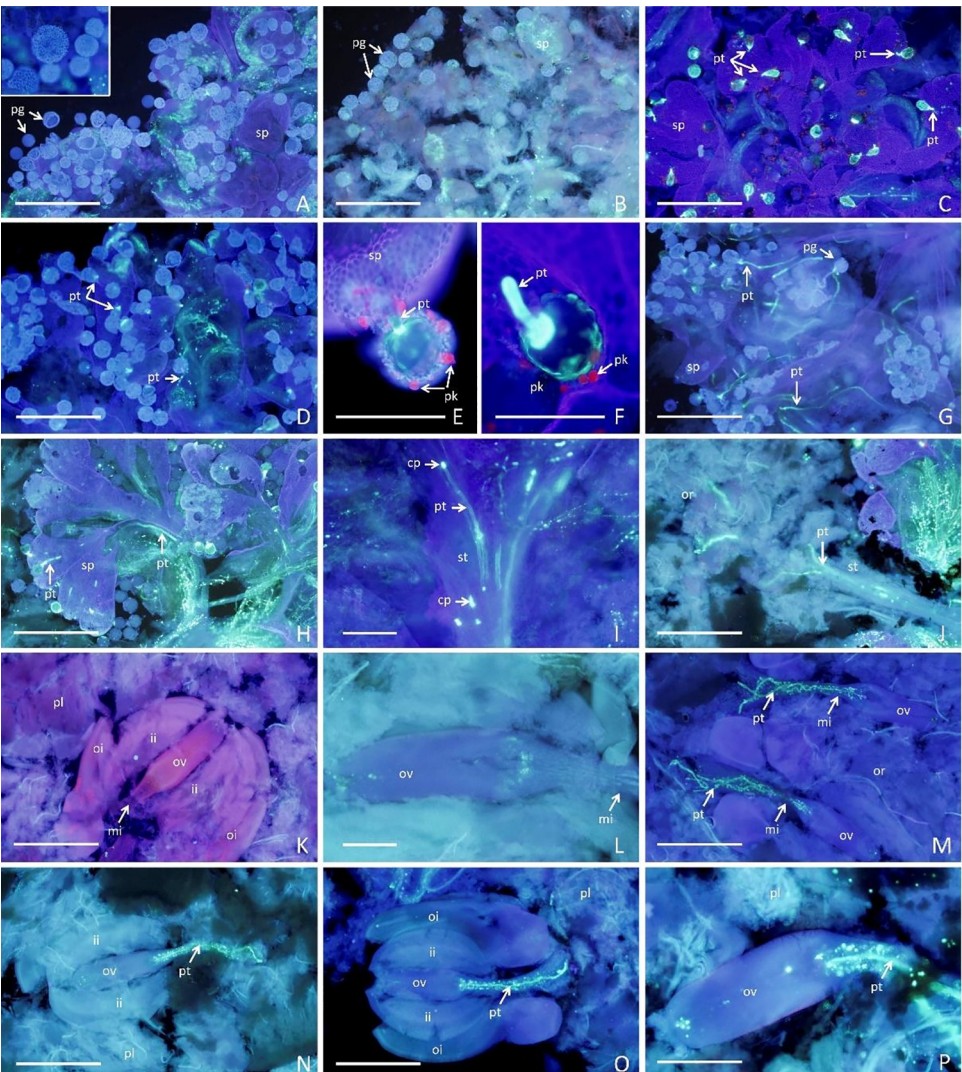

**Fig 4. Overview of cassava pistils stained with aniline blue and visualized by fluorescence microscopy 48 hours after pollination.** (A) Absence of germination and polymorphism in pollen grains from the BGM-0661 × BRS Jari cross; B) few pollen grains germinated on the surface of the stigmatic papillae with penetration of the pollen tube in the BGM-0661 × BGM-2167 cross; (C–D) pollen grains germinated on the surface of the stigmatic papillae with penetration of the pollen tube in the Aipim Abacate x BRS Novo Horizonte and BGM-0470 × BRS Novo Horizonte crosses, respectively; (E–F) presence of pollenkitt in the germinated pollen grains; (G) pollen tube growth ceased at the first third of the stylus in the 7909–04 × BGM-0661 cross; (H–I) pollen tubes within the stylet, showing the irregular deposition of callose plates; J) pollen tubes in the ovary with growth toward the ovary; (K–L) unfertilized ovule; (M–P) ovule being fertilized in compatible crosses (BGM-0728 × Aipim Abacate, BRS Novo Horizonte × Aipim Abacate, BGM1760 × BGM-2338, and BGM-2338 × BGM-0019, respectively). Bars: A–D, G–H, J–K, and M–P = 1 mm; E, F, I, and L = 200 μm.

F×M was highly significant for all traits, revealing that non-additive genetic effects (overdominance) explain a large part of the variance of these traits and can thus contribute to the efficient recombination of breeding populations (Table 4).

The weak relationship between female flower abortion rates × pollen viability, as well as the significant differences between the F×M for PGA, PGG, PTG, and NFO, provide evidence of pre-zygotic barriers among the 21 cassava clones. Only Aipim Abacate had high reproductive success, since it presented PTG near the ovary and high NFO (average: 2.11 eggs fertilized per female flower).

**Table 4. Likelihood ratio test for fixed effects and random effects (estimated via REML/BLUP for parent and female-male interaction (F×M) effects) on the number of pollen grains that adhered to the stigma surface (PGA), the number of pollen grains that germinated on the stigma surface (PGG), pollen tube growth (PTG), and the number of fertilized ovules (NFO) evaluated for crosses between 21 cassava parents.**

| Effect | Variation source | PGA | PGG | PTG | NFO |
|---|---|---|---|---|---|
| Fixed [W] | PGA | - | 27.25** | 100.40** | 36.81** |
| | Anthesis period | 6.74** | 1.84* | 12.61* | 4.33* |
| Random [T] | Parent | 33.73** | 6.74** | 2.01[ns] | 6.84** |
| | F×M | 51.49** | 84.05** | 65.25** | 71.13** |
| | Parent/F×M | 3.554 | 0.585 | 0.290 | 0.174 |
| Variance components | $\sigma^2_P$ | 0.30 | 0.06 | 0.25 | 0.04 |
| | $\sigma^2_{fxm}$ | 0.08 | 0.09 | 1.87 | 0.22 |
| | $\sigma^2_E$ | 0.50 | 0.43 | 3.98 | 0.90 |

[T] Random effects were significant at **, *, and [ns], denoting $p<0.001$, $p<0.05$, and non-significant likelihood ratio test results, respectively (evaluated via a $\chi^2$ test with one degree of freedom); [W] Fixed effects were significant at **, *, and [ns], denoting $p<0.001$, $p<0.05$, and non-significant results by the $X^2$ test.

F×M: Female-Male interaction effect.

The ratio between parent and F×M effects was highest for PGA (3.554), indicating a greater effect of parent on PGA (Table 4). However, this ratio was smaller for PGG, PTG, and NFO, of 0.585, 0.290 and 0.174, respectively, showing the predominance of the F×M effect for pollen-pistil interaction. The predominance of the F×M effect for PGG and PTG supports the hypothesis of SII in cassava for pollen-pistil interaction, considering that most parents had different PGG estimates and PTG locations in the crosses.

Since parents BGM-0685, BGM-1693, and BRS Dourada had SII indexes equal to zero, they were considered self-incompatible (Table 5). This result is expected for sporophytic SII, since no pollen grains were germinated on the surface of the stigma during self-pollination (Table 5). The genotypes BRS Kiriris, BGM-1760, and BRS Novo Horizonte were considered partially self-incompatible, with respective SII index values of 0.52, 0.70, and 0.78. Notably,

**Table 5. Self-incompatibility index (SII) and pollen tube growth (PTG) of self-pollinations performed among different cassava clones.**

| Self-pollination | Pollen tubes growth (%)[1] | | | | | | SII | SII evidence |
|---|---|---|---|---|---|---|---|---|
| | 0 | 1 | 2 | 3 | 4 | 5 | | |
| Aipim Abacate | - | - | - | - | - | 100.0 | 3.36 | Self-compatible |
| BGM-0019 | 28.5 | - | - | - | - | 71.5 | 4.05 | Self-compatible |
| BGM-0661 | 50.0 | - | - | - | - | 50.0 | 1.86 | Self-compatible |
| BGM-0685 | 100.0 | - | - | - | - | - | 0.00 | Self-incompatibility |
| BGM-0728 | 22.2 | - | - | - | - | 77.8 | 2.69 | Self-compatible |
| BGM-1693 | 100.0 | - | - | - | - | - | 0.00 | Self-incompatibility |
| BGM-1760 | 75.0 | - | - | - | - | 25.0 | 0.70 | Partially self-incompatibility |
| BGM-2020 | 25.0 | - | - | - | - | 75.0 | 1.50 | Self-compatible |
| BGM-2338 | 33.3 | - | - | - | - | 66.6 | 1.48 | Self-compatible |
| BRS Dourada | 100.0 | - | - | - | - | - | 0.00 | Self-incompatibility |
| BRS Jari | 25.0 | - | - | - | - | 75.0 | 3.08 | Self-compatible |
| BRS Kiriris | 88.8 | - | - | - | - | 11.2 | 0.52 | Partially self-incompatibility |
| BRS Novo Horizonte | 80.0 | - | - | - | - | 20.0 | 0.78 | Partially self-incompatibility |

[1] Scale: 0—pollen grains did not germinate on the surface of the stigma; 1—pollen grains germinated on the surface of the stigma; 2—tip of the pollen tube in the stylet; 3—tip of the pollen tube inside the ovary; 4—tip of the pollen tube close to the ovary; 5—pollen tube penetrated the micropyle.

most of the evaluated genotypes were classified as self-compatible. Therefore, the hypothesis regarding the presence of gametophytic and total sporophytic SII can be refuted based on progeny generation in 10 of the 13 self-pollinations observed in Experiment 2 (Table 5).

## Lack of association between population structure and crossing fertility

Three clusters were identified based on the genetic analysis of 86 cassava parents (used in Experiment 1) via the DAPC *find.clusters* function (Fig 5). Clusters 1, 2 and 3 included 30, 23 and 33 genotypes, respectively. The first 60 principal components (PC) retained >96% of the variance of the PCA, and two discriminating eigenvalues (LD) were maintained. LD1 separated the three clusters with no overlap. A heat map of the genomic relationship shows the parentage between the cassava genotypes belonging to the three groups, with some within-group subdivisions (Fig 5).

Although the DAPC analysis grouped the 86 parents with practically no overlapping genotypes in the three groups, there was no association between the genomic relationship of the parents versus the abortion and seed setting rates, especially in crosses between parents of the same cluster (i.e., 1x1, 2x2 and 3x3) (Fig 6). Between-cluster crosses (i.e., 1x2, 1x3 and 2x3) followed the same trend as within-cluster crosses, except for crosses between the parents of Clusters 1 and 3 (1x3), in which there was a significant and positive correlation between the

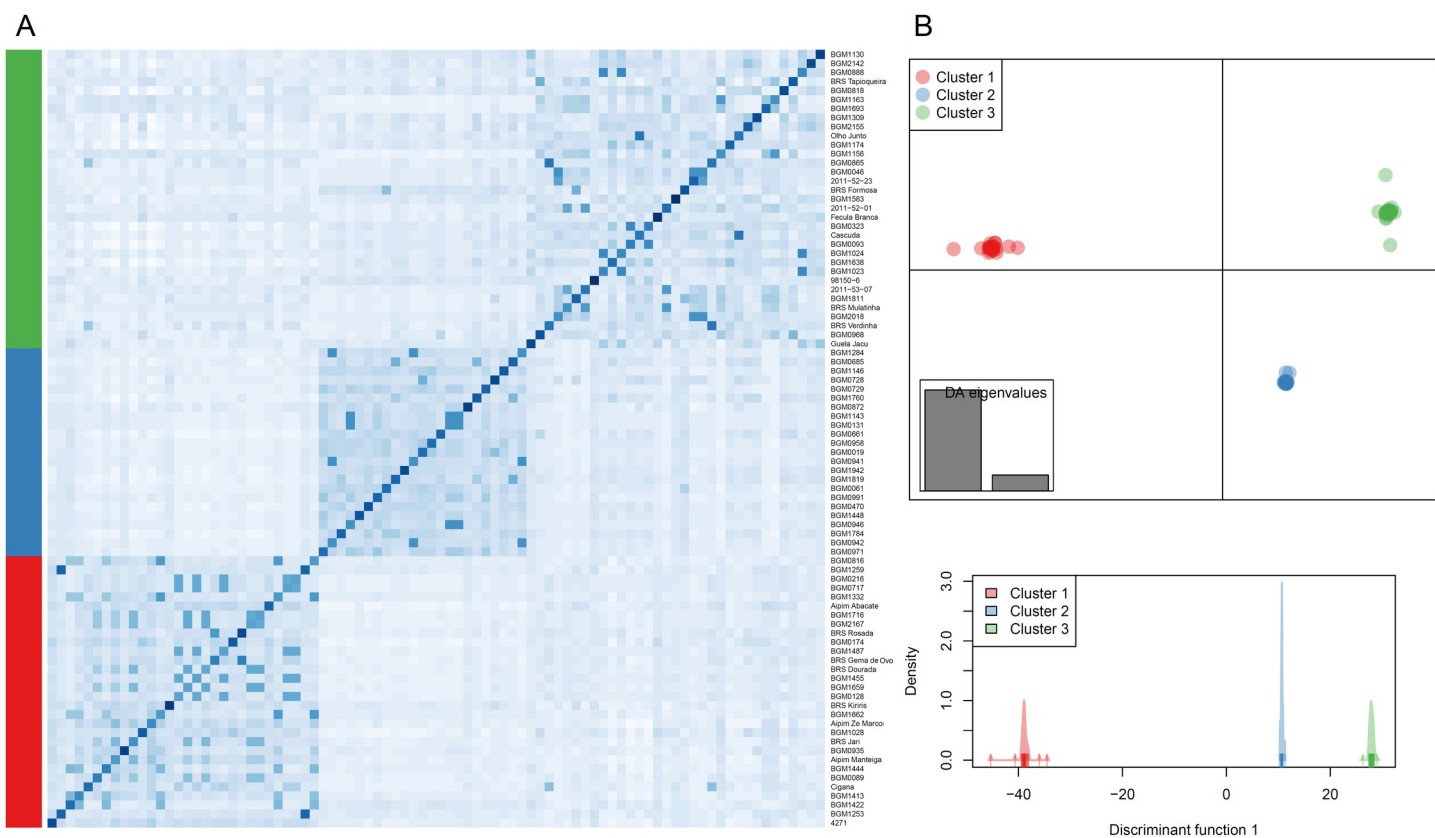

**Fig 5. Population structure and genomic relationship obtained via the analysis of 16,300 single-nucleotide polymorphisms among 86 cassava parents.** (A) Heat map of the genomic relationship matrix with the genotypes ordered according to the discriminant analysis of principal components (DAPC). The bar to the left of the heat map represents the DAPC group (i.e., Groups 1, 2 and 3 in red, blue and green, respectively). (B) The DAPC, in which the two axes of the upper graph represent the first two linear discriminants (LD). Each point represents an individual. The different subpopulations identified by the DAPC analysis are represented by the colors red, blue and green, while the bottom graph represents the distributions of the groups based on the first discriminant function only.

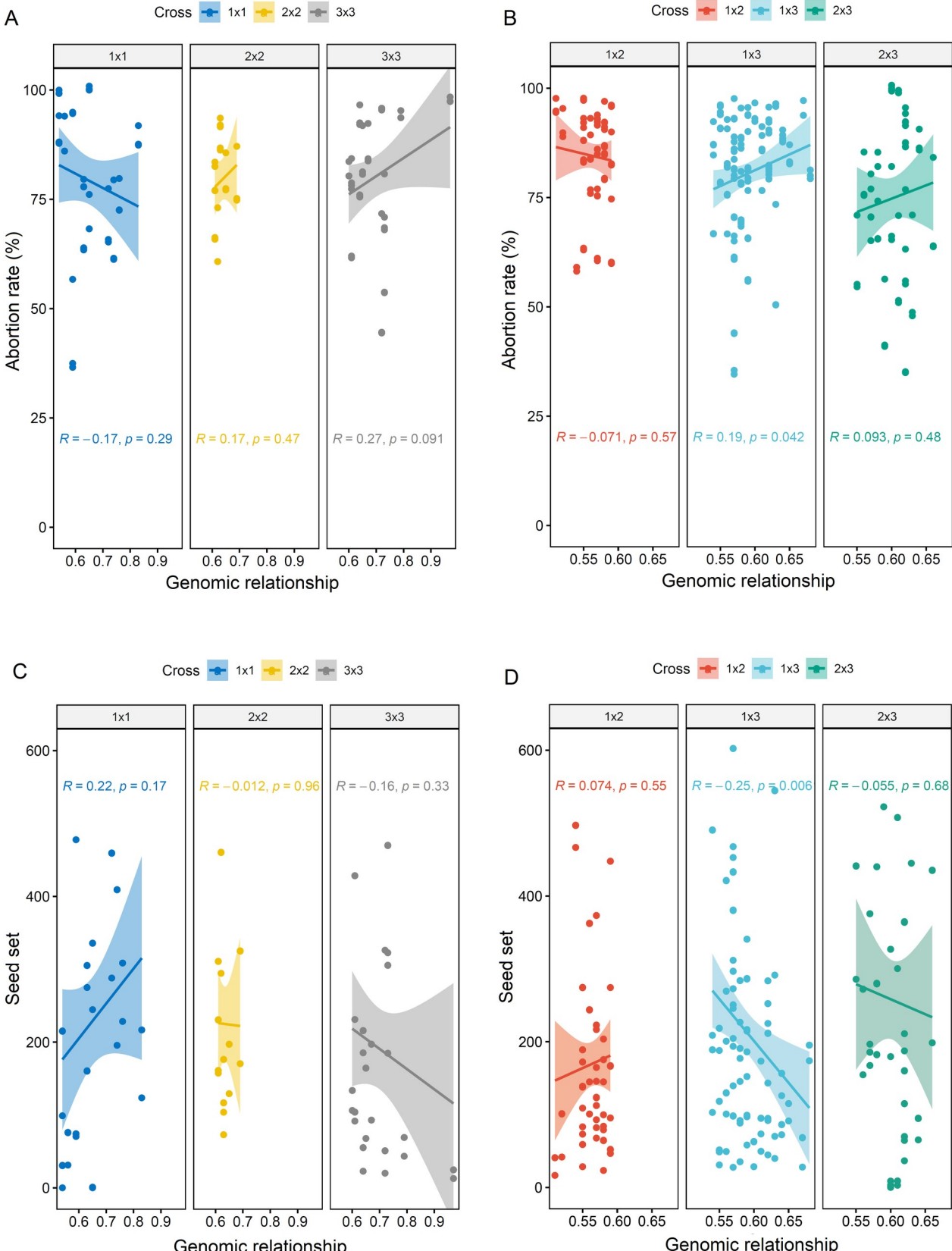

**Fig 6. Correlation between the genomic relationship calculated via the analysis of 16,300 single-nucleotide polymorphisms versus abortion rate and seed setting rate in cassava.** The 86 cassava parents were previously grouped into three clusters based on the discriminant analysis of principal components (DAPC), and crosses were performed between parents of the same cluster (i.e., 1x1, 2x2 and 3x3) or between clusters (i.e., 1x2,1x3 and 2x3).

genomic relationship of the parents and abortion rate (p = 0.042) as well as a negative correlation between the genomic relationship and seed setting rate (p = 0.006). However, the correlations had low magnitudes (0.19 and -0.25, respectively).

The estimates of parent and F×M effects of flowering traits of cassava (used in Experiment 2) were also grouped using the PCA method, which resulted in the formation of four heterotic groups with 9, 2, 6 and 15 genotypes (S9 Table). In principle, there was no relationship regarding the type of cassava genotypes based on breeding patterns, since the improved varieties and landraces were distributed in the three largest groups (Groups 1, 3 and 4).

Similar to the grouping performed based on molecular data, grouping based on parents and F×M did not generate crosses that led to enhancement of the process of fertilization and seed generation (S10 Table). Combinations between parents of the same group produced the highest numbers of pollen grains adhering to the surface of the stigma, increases in pollen tube growth, and the NFO—especially in crosses between parents of Groups 3 (3x3) and 4 (4x4). Therefore, crossing cassava clones within heterotic groups (based on parents and F×M) had a high probability of high reproductive success. Regarding crosses between parents of different groups based on parents and F×M, the groups of crosses 1×3 and 3x4 resulted in improvement of the reproductive process. These results corroborate the notion that the parents of Groups 3 and 4—when crossed with each other or between groups—tend to result in higher reproductive rates.

## Discussion

### Crossability in cassava

The successful development of new cassava varieties depends on combining useful alleles from different genotypes through the generation of segregating populations via artificial hybridization, followed by the phenotypic selection of superior $F_1$ plants in different progenies over various environments and years of cultivation. However, successful hybridization depends on the genotype and environmental conditions of cultivation, which generally present difficulties for flower synchronization between and within clones since cassava is a monoecious species with protogyny. In response to this challenge, some researchers have suggested storing pollen grains at low temperatures to perform crosses when female flowers appear [37].

Although such efforts aim to increase the reproductive success of cassava, our work demonstrates that the success rate of generating cassava progenies was generally low (≈29%) when using many crosses between clones with high genetic diversity. However, this percentage was higher than the success rates of 5% [13], 15.3% [14], and 15.6% [38] observed by other authors for cassava. Instead, it is comparable to the 38% reproductive success observed for crosses between cassava cultivars (*Manihot esculenta* subsp. *esculenta*) with the subspecies *M. esculenta* subsp. *flabellifolia* and *M. esculenta* subsp. *peruvian* [39].

Although reports have noted that cassava flowering varies widely with genotype, with some flowering early (approximately two months after planting) or late (two years after planting) [40], relatively few have noted high abortion rates and the complete failure to obtain progeny with certain crosses. While some authors have mentioned the lack of genetic barriers in experimental crosses between cassava genotypes [41], our experimental data indicate that the majority of crossings presented compatibility issues. For example, regarding the abortion rate, 5.9%

of the crossings exhibited 0–25% flower abortion, while 8.3% exhibited 26–50%, 34.9% exhibited 51–75%, and 50.9% exhibited 76–100% flower abortion.

Although there was no significant effect for the 10-day intervals factor, the interaction between the 10-day intervals × year of crossing was significant. Hence, uncontrolled abiotic effects over the different years affected the reproductive ability of cassava parents. Additionally, these results demonstrate: i) that some female parents have a strong ability to generate a large number of progenies due to their lower abortion rate (e.g., BGM-0728, 7909–4, BGM-0470 and BRS Tapioqueira, with 53.7%, 56.2%, 59.7% and 62.6% mean abortion rates, respectively); and ii) that both genetic and environmental factors affect cassava reproductive ability. Since the ovule or seed abortion is a consequence of hybrid sterility (post-zygotic) and crosses involving abortion of flowers due to pollen tube inhibition before fertilization as pre-fertilization barriers, we cannot accurately conclude whether the reproductive barriers observed were pre- or post-zygotic.

The present study also highlights the importance of both female and male effects on the abortion and seed setting rates of cassava. For this species, the male within female parent effect is higher, while female effects are predominant in other crops. For example, SII and female effect represented a significant barrier to sweet potato breeding, which results in wide variation in the specific combining ability effect between various direct and reciprocal crosses [42]. In sorghum, the impact caused by environmental stressors during pre-flowering in female parents was stronger than in males for several seed setting traits [43]. The parent effect in sorghum may be due to differences in genetic background, cytoplasmic male sterility systems, and corresponding fertility restorers that improve the development of $F_1$ hybrid varieties.

The fact that the female and male interaction is equally important for reproductive success in cassava highlights the importance of incorporating reproductive traits in breeding programs to improve recombination rates. Fortunately, the genetic variation of attributes related to flowering in cassava breeding programs is high and sufficient to improve the viability of crosses and increase the seed setting [6]. However, cassava breeders understand the great contrast related to seed production capacity in controlled pollination. For example, [13] reported wide variation in the reproductive efficiencies of different genotypes, particularly when used as female parents. Therefore, this represents a challenge that must be overcome when selecting parents.

## Effect of the environment on reproductive capacity

Due to the absence of significant differences between 10-day intervalss and the interaction between 10-day intervals and year of cultivation for the abortion of female flowers and seed setting, our results indicate that environmental factors contribute to a post-zygotic barrier in cassava. Temperature, rainfall and photoperiod are among the most important abiotic factors with capacity to influence reproductive ability in cassava [12]. We identified that high temperatures induce flower abortion and reduce the number of female flowers per inflorescence and seed setting. In turn, [12] observed that two cassava varieties (IBA980002 and TMEB419) exhibited stronger flowering inhibition under higher day and night temperatures (34 and 31°C, respectively) in comparison with milder temperatures (22–28°C during the day and 19–25°C at night). Additionally, in a recent study that aimed to determine the duration of stigma receptivity and pollen grain growth rate in cassava, [13] found that pollen tubes from tests conducted at higher average temperatures (33.3–33.7°C) grew slower than those from tests conducted at lower temperatures (29.5°C). These results support the hypothesis that environmental conditions affect the efficiency of sexual reproduction in cassava, and that appropriate planning of planting dates and locations can maximize seed production.

Notably, the production of cassava seeds was affected by environmental factors. The ideal temperature range for seed production varied between 22.5 and 24˚C, while temperatures above 25˚C reduced reproductive success of pollinations performed by 21%. Comparisons with other studies of cassava are difficult to make due to the scarcity of reports of this nature. However, temperature greatly influences reproductive success of numerous plant species. For example, excessively dry or cold periods can inhibit cocoa (*Theobroma cacao*) flowering in regions where seasonal variability in rainfall and temperature occur. Thus, environmental factors such as rainfall and temperature can have significant effects on flowering and subsequent fruit development [44].

Since the phenological response of cassava in terms of reproductive success depends on the environmental conditions of cultivation (i.e., the cumulative effects of temperature) and the limits of each genotype, we recommend that regional evaluations be conducted of the germplasm of various research centers to identify the accessions whose responses to crossing may be less susceptible to environmental factors within a standard range of normality in the ecoregion under analysis. Therefore, using historical climatic data together with knowledge about the effects of abiotic stresses on seed setting would allow breeders to optimize the recombination of parents when planting crossing blocks during periods with a higher likelihood of early and profuse flowering (i.e., May to September in the region of Cruz das Almas).

## Crossbreeding viability and reproductive success in cassava

The variability of pollen grain viability among cassava parents may explain the greater success of certain controlled pollinations. Microscopic analysis revealed certain abnormalities during fertilization, such as reticulated deposition of callose in the pollen tube, pollen tube growth cessation in a certain region of the stylet, and poor germination of pollen grains. Previous studies have shown a positive correlation between callose buffer deposition and pollen tube growth [45]. Differences in callose deposition have also been described in interspecific and intergeneric crosses in Bromeliaceae, in which they are directly related to reproductive success in controlled crosses [18]. In tomato, poor deposition of callose plates is a consequence of slower pollen tube growth in plants with reduced pollen receptor kinase (*LePRK2*) expression [46]. Moreover, reports have indicated that callose on the pollen tube wall plays an important role in maintaining the osmotic balance of the pollen tube [47]. According to [48], callose deposition helps to determine the incompatibility system, gametophytic competition, and viability of the stigma and ovule. Future investigations should aim to determine the effects of the irregular arrangement of callose plates on cassava crosses.

Regarding pollenkitt, all genotypes presented this substance on the surfaces of pollen grains. Generally, pollen grains with a large amount of pollenkitt are more likely to fertilize ovules since pollenkitt is composed of glycoproteins responsible for the recognition of pollen-stigma compatibility. Pollenkitt also aids the fertilization process by facilitating the adhesion of pollen grains to the stigma in the exine cavities, protecting the grains against water loss, ultraviolet radiation, and hydrolysis by extracellular enzymes and rehydrating pollen grains [49]. The oily component of pollenkitt preserves and protects recognition glycoproteins during their transport and attachment to the stigma surface, which may explain our observation of greater adherence to stigmas of some genotypes.

In the second part of the study, a reproductive success rate of approximately 32% was observed among crossings with parents selected in the first stage of the research by using male flowers collected in anthesis. However, according to [20], the viability of cassava pollen in immature flowers (3.2 mm in diameter) is high (>80%), decreases gradually with male flower maturity, and reaches approximately 10% during anthesis. Therefore, although controlled

 

pollinations have standardized the collection of pollen grains from male flowers during anthesis, using pollen grains from more immature flowers may result in greater success rates for cassava crosses, and thus deserves further investigation.

In most cases of pollen grain germination, at least one ovule is fertilized in a flower 48 hours after pollination (PAH), which—in principle—is the time required for pollen tubes to reach the embryo sac. The present study revealed a delay in the arrival of pollen tubes to the embryonic sac. This could be related to less pollen vigor or incompatibility, but this does not necessarily prevent fertilization. Similar to the observation of Ramos *et al.* [13], the long period required for the pollen tube to reach the embryonic sac varied widely between crosses. The same authors also reported that pollen tubes reached the embryonic sac at approximately 26 PAH in 2% of the samples, with a significant increase in the proportion of pollen tubes reaching the embryonic sac at 74 PAH (27%). Therefore, these results confirm the slow growth of pollen tubes, especially from micropyles.

In the present study, the average pollen viability was 60.10% (range: 13.33–96.66%). Therefore, unlike [20], our findings highlight the existence of significant differences in pollen grain viability between the cassava genotypes, which were determined using Alexander's method. This information is important when selecting parents for use in crossing blocks to maximize seed production. The use of a genotype with low pollen viability could only be justified if it has uniquely desirable traits and high agronomic value (e.g., clone 7909–04). One factor that helps to explain low pollen viability is the size of pollen grains, since grains of normal to large size have viability >80%, while the viability of small pollen grains can vary between 8 and 29% [20]. The form of pollen grain dimorphism and trimorphism observed in 90% of the analyzed genotypes has also been reported in other studies of cassava [20, 50]. Other factors that help to explain pollen viability include pollen age and environmental variables (e.g., temperature and humidity) [51]. In the Embrapa cassava breeding program, low seed yield due to controlled pollination and seed viability has been reported as a problem to be solved. Notably, pollen grain sterility can contribute to a low seed setting of this species.

In cassava, the stigma receptivity at post-anthesis (20–24 hours after anthesis) was slightly higher than during pre-anthesis and anthesis. According to [13], cassava stigmas remain receptive for up to three days after anthesis, after which the stigmas typically fall from the pistils. On the other hand, [20] reported a negative correlation between the viability and frequency of small pollen grains and also noted that the gradual death of pollen grains occurs with their maturity, resulting in the reduction of pollen viability at anthesis. Based on the mentioned results of stigma receptivity, we recommend that pollinations should continue until one day after anthesis.

Sexual reproduction in cassava is generally quite variable and relatively inefficient, since only 29–31% of the controlled crosses conducted in the present study were fertilized. In other more extreme situations, fruit production from open (uncontrolled) pollination was only 5% of what was expected [13]. These authors also reported the following: i) great genetic variation in the reproductive capacity of different parents, especially when used as a female parent; and ii) no difference in reproductive efficiency between self-pollination and cross-pollination.

Despite applying a standardized procedure for deposition of pollen grains during controlled pollinations, there was wide variation in the number of pollen grains adhered to the stigma surface among male parents. This can be explained because the stigmatic cuticle ruptures only after anthesis, which explains the higher germination rate of pollen at that stage. However, this needs to be verified by histochemical studies. Another hypothesis is that the stigmas of cassava are considered dry, with no exudates for the adherence of pollen grains. In species with dry stigmas (e.g., *Arabidopsis thaliana*), this interaction is highly selective and only allows the adherence of a limited set of pollen grains [52]. Generally, the adhesion of pollen grains occurs

 

in seconds, is extremely strong, and is mediated by substances present in the exine (e.g., pollenkitt) [53], and glycoproteins, such GRP17 as in *Arabidopsis thaliana* [54]. Additional studies of cassava are required to provide a better understanding of the molecular components responsible for pollen grain adhesion to the floral stigma.

## Matching ability between cassava parents

The selection of parents to improve cassava flowering may not exclusively depend on their performance *per se*, but rather on their ability to combine with other parents and generate transgressive individuals. Therefore, female-male interaction should be an important criterion that plant breeders can evaluate to compare the reproductive performance of clones and determine the genetic effects in order to improve recombination of breeding populations. Detailed genetic studies of the female-male interaction are required to improve estimates of reproductive capacity.

In the case of cassava flowering, the genetic effects of parent and F×M were significant for number of pollen grains adhering to the stigma surface, number of germinated pollen grains, and the number of fertilized ovules. Notably, only F×M significantly affected the place where the pollen tubes stop growing. Generally, the parent effect at flowering traits is an important indicator of their potential to generate large progenies, allowing to breeders to optimize the cross-pollination activity.

Although significance was observed for parent effect among most traits, the effects of F×M were more pronounced, corroborating the hypothesis of presence of SII in cassava. Therefore, the proportion of aborted ovules is genetically controlled and varies widely among cassava clones. To the best of our knowledge, this is the first study to address this topic in cassava for flowering attributes. However, a thorough analysis of the genetic causes of heterosis for these traits is beyond the scope of this work.

The success of breeding programs for certain traits depends on the variability of populations and the extent to which important traits are heritable. Therefore, knowledge of the genetic architecture of attributes associated with cassava flowering can help to formulate more effective breeding strategies to increase the number of individuals in segregating populations. In particular, reducing the abortion rate could improve the recombination rates of cassava parents in cassava breeding programs.

## Partial self-incompatibility in cassava

Some indications of pre-zygotic barriers among the 21 evaluated cassava clones included: i) significant differences in F×M between the traits PGA, PGG, PTG and NFO; ii) a weak relationship between the abortion rates of flower × pollen viability; and iii) the presence of total and partial SII in self-pollination. The limited number of evaluated parents makes it difficult to estimate the number of genes and alleles involved in expressing the observed incompatibility. Performing a large number of controlled crosses within and between families of full-sibs can help to identify and map genes involved in the reproductive mechanisms of cassava.

Since one of the main challenges in cassava breeding is the low seed production from controlled crosses and self-pollinations, we considered SII, which often results in little or no seed production after self-pollination. This typically occurs due to gene products from the locus that prevent germination and/or pollen tube growth whenever the male (pollen) and female (pistil) tissues express the cognate S-alleles. This mechanism has evolved to prevent the occurrence of inbreeding in angiosperms, which often results in the considerable loss of agronomic performance [3, 4, 55].

Our results indicate that most of the evaluated genotypes were classified as self-compatible, which helps to refute the hypotheses of gametophytic and total sporophytic SII. However, a single SII system in cassava is unlikely, since three cassava genotypes were classified as self-incompatible (SII = 0), while three others were considered partially self-incompatible (SII range: 0.52–0.78). These results contrast with the study by [13], who observed no significant differences in pollen tube growth or the density of pollen grains that adhered to the stigma surface, regardless of the pollination method (self-pollination versus cross-pollination). Notably, this result suggests there are no barriers to the realization of self-pollination in cassava. Therefore, future studies of pollen-pistil interaction should be conducted with a greater number of genotypes and progenies to verify which factors led to inconsistencies in the results of the present study. Understanding complete or partial self-compatibility in cassava and conducting refined studies of the segregation of S locus aimed at identifying associated molecular markers are of paramount importance to increase the efficiency of parental recombination in crossing blocks.

Previous molecular studies of the S locus have demonstrated—depending on the nature of the S haplotypes expressed in the pollen and pistil—the existence of different compatibility relationships. In *Solanum* species, the S locus has high allelic diversity, with an amino acid sequence similarity ranging from 32.9 to 94.5% among the identified 16 S alleles, despite the small number of genotypes evaluated [56]. Additionally, there is evidence of a quantitative nature involved in SII segregation. In citrus, SII is likely controlled by an S-RNase and several SLF genes that act with S determinants [57]. Notably, the aforementioned studies provide pathways for future research aimed at a better understanding of the self-compatibility phenomenon in cassava.

In predominantly self-compatible species, high rates of fruit abortion may be due to deleterious genes in the population [58]. Since cassava is a vegetatively propagated species, the accumulation of deleterious mutations is expected due to limited recombination and a domestication bottleneck. [59] observed that hybrid cassava cultivars have a higher proportion of heterozygosity in relation to their parents, despite their reduced genetic diversity, which demonstrates that deleterious mutations are maintained.

## Molecular and phenotypic clustering versus reproductive success in cassava

The grouping of cassava parents based on molecular data and parent/F×M for attributes associated with fertilization and seed setting did not increase reproductive viability when analyzing between-cluster crosses. Therefore, the genomic relationship of cassava parents seems to have little influence on the attributes associated with fertilization and seed formation. Other authors have reported that compatibility in cassava is high, even under self-fertilization [13].

The formation of contrasting groups to maximize heterotic effects in cassava can assist in recombination between elite parents. This increases the probability of breaking the linkage between traits to obtain high genetic gain and greater seed production in crossing blocks. However, our results demonstrate that a few significant and low-magnitude correlations explained the relationship between the genomes of parents and abortion and seed setting rates. According to [60], in common wheat, genotype clusters based on molecular data (genetic distance—GD) and the subsequent analysis of heterotic effects from a cross between clusters did not result in strong significant correlations between GD and high parent heterosis for 1000-grain weight, spikelet number, harvested spikes, and yield [60]. Thus, our results suggest that the selection of crosses for the greatest chance of success in generating cassava seeds should only consider the flowering capacity of the parent and its combining ability with other parents.

## Final remarks

Despite advances in several genomic approaches and the high-throughput screening of cassava, many gaps remain in basic research areas (e.g., reproductive biology). Improvement of cassava seed setting is of fundamental importance for farmers to choose the main agricultural inputs, which are cassava varieties. Only with advanced technological packages and good cultivars will it be possible to reach the maximum root and starch yield.

Several techniques can be used to overcome reproductive barriers in cassava, such as the pollination of cut stylets, grafting of stylets, and placental pollination. Thus, it is critical to identify the time and location of these barriers and use appropriate and efficient techniques to overcome them. Notably, some genotypes already have low pollen viability as a barrier to cross-pollination.

In the present study, pre-zygotic barriers that led to a high rate of fruit abortion were demonstrated. For example, we observed a genetic effect among female parents, the influence of environmental factors (e.g., temperature and rainfall), variable pollen viability, pollen grain polymorphism, and evidence of partial SII.

## Supporting information

**S1 Fig. Boxplot of progeny for abortion rate (%) and seed setting rate in Experiment 1.**
(TIF)

**S2 Fig. Relationship between abortion rate (%) and seed setting rate based on the average of cassava progenies from Experiment 1.**
(TIF)

**S1 Table. Genotype, origin, plant shape, cyanogenic compounds (HCNs), and root color traits of the 91 cassava genotypes included in Experiments 1 and 2.**
(DOCX)

**S2 Table. Likelihood ratio test for fixed effects and random effects for seed setting and abortion rates in cassava progenies (obtained from 2016 to 2018).**
(DOCX)

**S3 Table. Assessment of the self-pollination performed in 13 of the 21 clones (Experiment 2) to determine the number of pollen grains that adhered to the surface of the stigma (PGA), the number of pollen grains that germinated on the surface of the stigma (PGG), pollen tube growth in the pistil (PTG), and the number of fertilized ovules (NFO).**
(DOCX)

**S4 Table. Best linear unbiased predictors for the random effects of female and male parents nested to female parents for seed setting characteristics, abortion rate, and seed setting rate of selected parents included in Experiment 2.**
(DOCX)

**S5 Table. Number of fertilized flowers for each parent included in Experiments 1 and 2.**
(DOCX)

**S6 Table. Relationship between the number of pollen grains that adhered to the stigma surface (PGA) and pollen tube growth in the pistil (PTG) with the number of pollen grains that germinated on the stigma surface (PGG) during pollen tube development in the pistil of different cassava crosses.**
(DOCX)

**S7 Table. Frequency of number of pollen grains that adhered to the stigma surface (PGA), number of pollen grains that germinated on the stigma surface (PGG), pollen tube growth in the pistil (PTG) and number of fertilized ovules (NFO) according to observation of the pollen tube in the cassava pistil at pre-anthesis, anthesis, and post-anthesis.**
(DOCX)

**S8 Table. Mean test of anthesis periods and the number of pollen grains that adhered to the stigma surface (PGA), the number of pollen grains that germinated on the stigma surface (PGG), pollen tube growth (PTG), and the number of fertilized ovules (NFO) evaluated for crosses between 21 cassava parents.**
(DOCX)

**S9 Table. Heterotic groups based on principal component analysis of the parent and female-male interaction effect (parent/F×M) for attributes related to the reproductive abilities of different cassava genotypes.**
(DOCX)

**S10 Table. Frequency observed of number of pollen grains that adhered to the stigma surface (PGA), number of pollen grains that germinated on the stigma surface (PGG), pollen tube growth in the pistil (PTG) and number of fertilized ovules (NOF) between genotypes clustered based on parent and Female-male interaction (Parent/F×M) in cassava.**
(DOCX)

## Author Contributions

**Conceptualization:** Massaine Bandeira e Sousa, Luciano Rogerio Braatz de Andrade, Eder Jorge de Oliveira.

**Data curation:** Massaine Bandeira e Sousa, Luciano Rogerio Braatz de Andrade, Everton Hilo de Souza.

**Formal analysis:** Massaine Bandeira e Sousa, Luciano Rogerio Braatz de Andrade, Everton Hilo de Souza.

**Funding acquisition:** Alfredo Augusto Cunha Alves, Eder Jorge de Oliveira.

**Investigation:** Massaine Bandeira e Sousa, Luciano Rogerio Braatz de Andrade, Everton Hilo de Souza.

**Methodology:** Massaine Bandeira e Sousa, Luciano Rogerio Braatz de Andrade, Everton Hilo de Souza.

**Project administration:** Eder Jorge de Oliveira.

**Resources:** Eder Jorge de Oliveira.

**Supervision:** Eder Jorge de Oliveira.

**Writing – original draft:** Massaine Bandeira e Sousa, Luciano Rogerio Braatz de Andrade, Everton Hilo de Souza.

**Writing – review & editing:** Alfredo Augusto Cunha Alves, Eder Jorge de Oliveira.

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
