## [Decision Letter · Decision Letter 0]

17 Sep 2021

PONE-D-21-16565Reproductive barriers in cassava: Factors and implications for genetic improvementPLOS ONE

Dear Dr. Oliveira,

First of all, I will like to apologise for delay, one of the reviewer was not feeling well and took longer than expected. Sending MS to a new reviewer would have taken even longer, so I decided to give him more time. Thank you for submitting your manuscript to PLOS ONE. After careful consideration, we feel that it has merit but does not fully meet PLOS ONE’s publication criteria as it currently stands. Therefore, we invite you to submit a revised version of the manuscript that addresses the points raised during the review process.

The manuscript is a good piece of work but both reviewers have certain reservation and will like to revise some of the portions from methodology and portion on SI. I hope you will find reviewer comments helpful and for the betterment of Manuscript. 

We look forward to receiving your revised manuscript.

Kind regards,

Shailendra Goel, Ph.D.

Academic Editor

PLOS ONE

Journal Requirements:

“The authors thank Conselho Nacional de Desenvolvimento Científico e Tecnológico, Fundação de Amparo à Pesquisa do Estado da Bahia, and Coordenação de Aperfeiçoamento de Pessoal de Nível Superior for financial support. This work was also supported by the NEXTGEN Cassava project, through a grant to Cornell University by the UK’s Foreign, Commonwealth & Development Office (FCDO) and the Bill & Melinda Gates Foundation (Grant INV-007637 http://www.gatesfoundation.org).”

“●         Eder Jorge de Oliveira: CNPq (Conselho Nacional de Desenvolvimento Científico e Tecnológico). Grant number: 409229/2018-0, 442050/2019-4 and 303912/2018-9

●          Eder Jorge de Oliveira: FAPESB (Fundação de Amparo à Pesquisa do Estado da Bahia). Grant number: Pronem 15/2014

●          Everton Hilo de Souza: CAPES (Coordenação de Aperfeiçoamento de Pessoal de Nível Superior)

●          Eder Jorge de Oliveira, Alfredo Augusto Cunha Alves, Massaine Bandeira e Sousa and Luciano Rogerio Braatz de Andrade: UK’s Foreign, Commonwealth & Development Office (FCDO) and the Bill & Melinda Gates Foundation. Grant number: INV-007637

●          The funder provided support in the form of fellowship and funds for the research, but did not have any additional role in the study design, data collection and analysis, decision to publish, or preparation of the manuscript.”

Reviewers' comments:

Reviewer's Responses to Questions

**Comments to the Author**

1. Is the manuscript technically sound, and do the data support the conclusions?

Reviewer #1: Partly

Reviewer #2: Yes

2. Has the statistical analysis been performed appropriately and rigorously? 

Reviewer #1: Yes

Reviewer #2: Yes

3. Have the authors made all data underlying the findings in their manuscript fully available?

Reviewer #1: Yes

Reviewer #2: Yes

4. Is the manuscript presented in an intelligible fashion and written in standard English?

Reviewer #1: Yes

Reviewer #2: No

5. Review Comments to the Author

Reviewer #1: Comments: PONE-D-21-16565.pdf

The MS is based on the findings of an extensive work on identifying the crossability barriers between the cultivars/landraces of Cassava. The study is important for genetic improvement of crop and heterotic breeding. Baseline data has been generated for the reproductive attributes, especially pertaining to those required for performing successful crosses. The authors have used pertinent statistical analysis as expected from any well-trained agronomists. However, there are several issues with the MS at this stage. For example, although the authors may be familiar with the system, the readers are not. It would be important to provide some information on the structure of flower, sexuality in the plant species and if tis naturally pollinated by some particular group of pollinators. This information helps in understanding that how the species has evolved its reproductive strategy in nature. Then comes the issue of floral biology. The MS lack information on the extent on pollen production among cultivars. It would be pertinent to use standard reproductive biological methods which I have specified below.

The language is style is acceptable, but the MS can be shortened further. The MS can be modified keeping in view some of the specific points raised below.

Specific comments are mentioned below and some are annotated in the attached pdf of the MS

Line 51: What does the term botanical seed refers to? Also Use of the term style would be better

Line 79: Does the family here means the accessions? And does fertility here means hybrid fertility?

Line 118-119: How long the pollinated flowers were bagged? Did bagging caused senescence of treated flowers?

Line 127: What does the environmental mean here? Did different times included different plants as well or the same plant.

Line 169: Did you use decolorized aniline blue? Need a reference here.

Lines 173-183: How was the receptivity of stigma tested? There are methods to ascertain the duration and peak time of stigma receptivity, by methods such as peroxidase test or by localizing non-specific esterases. the most receptive test then can be used for ascertaining pre-fertilization crossability barriers. Without identifying the receptive stage or as the method described here would obviously give different reading at different stages of the

flower. Stigma-specific incompatibility barriers are often stage-specific, as in Brassicas.

Line 183: What is a pseudomicropyle? need clarity here.

Lines 184-190: Alexander's stain is not the recommended method to ascertain viability of pollen for the reason that this stain as the title of the reference also suggests is a method to differentiate abortive (sterile) from non abortive (fertile) pollen.

Alexander MP (1969): DIFFERENTIAL STAINING OF ABORTED AND NONABORTED POLLEN. Stain Technology 44:117-122.

Viability test should include use of vital stain like TTC or FDA which tells if the pollen is alive or not at that particular time or how many (%) pollen are viable in a sample. Alexander stain will give results even with fixed pollen samples because it stains the cytoplasm and does not give indication of the activity of enzymes.

Line 212: Was there any particular reason for this modification in percentages or addition of PVP? It may be mentioned here.

Line 245: while there may be reproductive barriers among the accessions crossed in the experiment, the outcome would have been different in had the pollen viability and stigma receptivity test been done in accordance with the standard methods. Wrong or no assessment of viability has been one of the reasons of wrong judgements/conclusion in the past (see reference below)

American Journal of Botany

Vol. 82, No. 9 (Sep., 1995), pp. 1186-1197 (12 pages)

Line 279: Please see my comments above.

Line 339:...further growth of the tube in the stylar region.

Line 341-342: Not clear. Does this figure refer to 0.38% of the crosses? if so then it is obvious that pollen tubes have failed to reach the ovules.

Line 354: In manual deposition of pollen the PGG values are likely to be high. How the adherence was tested. Did the authors wash the stigma after certain duration of pollen deposition? Among incompatible crosses most of the pollen will not adhere and would be removed in washing.

Line 417-423: Did the cultivars varied in terms of outcomes of unilateral vs bilateral crossings?

Line 432: In my opinion, the data generated form breeding experiments is based on some flawed methods on pollen viability and not selecting a suitable stigmatic stage for crossing experiments.

Line 470: Ovule or seed abortion should be categorised as the consequence of hybrid sterility (post-zygotic) and crosses involving abortion of flowers due to pollen tube inhibition before syngamy as prefertilization barriers.

Line 516: Stylar inhibition of pollen tubes would mean a gametophytic control rather than sporophytic one.

Line 519: Poor callose plug deposition due to incompatible reaction is consequence rather than a cause of slow tube growth rate manifested, which in turn is resulted from incompatible reaction in the pistil

Lines 539-541: SLower pollen tube growth rate in the pistil in such crosses may be due to several reasons including pollen vigour, incompatibility reaction. pollen grains with moderate viability also grow slowly and then fail to reach the pistil while the vigorous ones

grow faster.

Line 555: Do you mean to say hybrid sterility or seed viability?

Lines 557-564: Why this result has been discussed in the context of receptivity so late in the MS. It should be mentioned right from the Methods onwards that in vivo test was done to test the receptivity.

Line 572: It means that the stigma has cuticle pellicle layer that is ruptured only post anthetically, which explains the higher germination rate of pollen at those stages. However, This needs to be verified histochemically.

Line 575: ...and glycoproteins such GRP17 as in Arabidopsis thaliana and arabinogalactans.

Line 601: Is there any information on the allelic diversity at the SI locus in Cassava. If there are many alleles, then there could be dominance relationship in play effecting diverse outcomes on fertilization success and seed set.

Reviewer #2: The manuscript describes a work which is not very commonly pursued. The experiments were designed meticulously to test the hypotheses. However, the MS needs to be improved significantly for its English language and expressions so that it is comprehensible to a reader without any ambiguity and without any need for speculation by the reader.

e.g. Line 435- "Combinations between parents of the same group produced the highest numbers of pollen grains adhering to the surface of the stigma"- Here most probably the authors want to state that when crosses were made between parents of the same group, a higher number of pollen grains remained adhered to the surface of stigma.

Therefore, the English of the MS needs to be improved and made more easily comprehensible. Further, following are my comments which the authors may clarify and incorporate in the MS

1. Line 242- "Six progenies from crosses between 10 different parents had a 100% abortion rate". Do you mean six cross combinations that involved 10 genotypes as

2. Line 295- How do you distinguish between pollen viability and pollen germination?

3. Line 160- "After selecting these genotypes, another 135 random 161 cross combinations were performed using 1,037 female flowers". Do you mean controlled crosses for 135 parental combinations were made? In that case, you may like to remove the word, "random".

4. Line 182- "4) growth of the pollen tube close to the ovary". Do you mean ovary or ovule in this sentence?

5. What was the depth threshold for calling the GBS-SNPs during the TASSEL pipeline?

6. Did the authors find any set of male parents with which the seed set was significantly higher than rest of the males. Did these male parents belonged to the same group based on SNP data?

7. Line 489- Is the data on relative humidity available for Expt 1? Was there any correlation of seed set rate and level of humidity?

6. PLOS authors have the option to publish the peer review history of their article (what does this mean?). If published, this will include your full peer review and any attached files.

Reviewer #1: No

Reviewer #2: No

---

## [Author Response · Author response to Decision Letter 0]

8 Oct 2021

Reviewer #1: Comments: PONE-D-21-16565.pdf

The MS is based on the findings of an extensive work on identifying the crossability barriers between the cultivars/landraces of Cassava. The study is important for genetic improvement of crop and heterotic breeding. Baseline data has been generated for the reproductive attributes, especially pertaining to those required for performing successful crosses. The authors have used pertinent statistical analysis as expected from any well-trained agronomists. However, there are several issues with the MS at this stage. For example, although the authors may be familiar with the system, the readers are not. It would be important to provide some information on the structure of flower, sexuality in the plant species and if tis naturally pollinated by some particular group of pollinators. This information helps in understanding that how the species has evolved its reproductive strategy in nature. Then comes the issue of floral biology. The MS lack information on the extent on pollen production among cultivars. It would be pertinent to use standard reproductive biological methods which I have specified below. The language is style is acceptable, but the MS can be shortened further. The MS can be modified keeping in view some of the specific points raised below. 

1) Line 51: What does the term botanical seed refers to? Also Use of the term style would be better

Response: In commercial systems, cassava is vegetatively propagated using 15-30 cm portions of the mature stem. The stem cuttings are sometimes referred to as “stakes” or “seeds”. We use “botanical seeds” to refer to true seeds, where in the cassava crop, these are used only for breeding purposes.

We added the following information in the text “….obtained by sexual reproduction…”

2) Line 79: Does the family here means the accessions? And does fertility here mean hybrid fertility?

Response: The family means a progeny from crosses between two genotypes (controlled crosses) or from self-fertilization. 

Yes, fertility means hybrid fertility, which is the crossability among cassava varieties in terms of seed setting capability and germinability of the hybrid seeds. We incorporated “hybrid fertility” in the manuscript to improve clarity. 

Line 118-119: How long the pollinated flowers were bagged? Did bagging caused senescence of treated flowers? 

Response: This answer can be found at the end of the same paragraph, namely “The protective bag covered the inflorescence until the seeds were released and collected, which occurred approximately 2 to 3 months after pollination”. 

There was no evidence that bagging caused senescence of pollinated flowers. Flowers are protected with bags to avoid insect damage or even uncontrolled pollination, and to keep the seeds within the bags due to natural dehiscence of cassava seeds. 

Line 127: What does the environmental mean here? Did different times included different plants as well or the same plant. 

Response: The environmental effect is mentioned at the beginning of the paragraph, as uncontrolled field effects in 10-day intervals, which could be differences in pluviosity, temperature, solar radiation and other climatic effects. The different times or environments (10-day intervals) included the same plants. 

Line 169: Did you use decolorized aniline blue? Need a reference here.

Response: Yes, we used decolorized aniline blue and we also added two references in the text to clarify the statement.

 Franklin W. Martin (1959) Staining and Observing Pollen Tubes in the Style by Means of Fluorescence, Stain Technology, 34:3, 125-128, DOI: 10.3109/10520295909114663

 Souza EH, de Versieux LM, Souza FVD, Rossi ML, Costa MAPC, Martinelli AP. Interspecific and intergeneric hybridization in Bromeliaceae and their relationships to breeding systems. Scientia Horticulturae. 2017; 223(15): 53-61.

Lines 173-183: How was the receptivity of stigma tested? There are methods to ascertain the duration and peak time of stigma receptivity, by methods such as peroxidase test or by localizing non-specific esterases. the most receptive test then can be used for ascertaining pre-fertilization crossability barriers. Without identifying the receptive stage or as the method described here would obviously give different reading at different stages of the flower. Stigma-specific incompatibility barriers are often stage-specific, as in Brassicas.

Response: We used in vivo tests to analyze the stigma receptivity. We made some modifications in the methodology to clarify it. We rewrote the sentence as:

“Stigma receptivity was analyzed in vivo considering three separate anthesis periods: pre-anthesis; anthesis; and post-anthesis. Three to five replicates (i.e., flowers) were evaluated for each parent combination and anthesis period. Pre-anthesis was defined as almost mature flowers (one day before anthesis), anthesis as open and functional flowers (up to 4 hours after opening), and post-anthesis as fully open flowers evaluated one day after anthesis”. 

Moreover, in cassava, the literature shows that stigmas remain receptive for up to three days after anthesis (doi: 10.1080/19420889.2019.1631110).

Line 183: What is a pseudomicropyle? need clarity here. 

Response: The correct word is “micropyle”.

Lines 184-190: Alexander's stain is not the recommended method to ascertain viability of pollen for the reason that this stain as the title of the reference also suggests is a method to differentiate abortive (sterile) from non abortive (fertile) pollen. 

Alexander MP (1969): DIFFERENTIAL STAINING OF ABORTED AND NONABORTED POLLEN. Stain Technology 44:117-122. 

Viability test should include use of vital stain like TTC or FDA which tells if the pollen is alive or not at that particular time or how many (%) pollen are viable in a sample. Alexander stain will give results even with fixed pollen samples because it stains the cytoplasm and does not give indication of the activity of enzymes.

Response: We believe that use of Alexander's solution is a standard method employed in several pollen viability studies, whose results are very similar for the in vitro pollen germination in several species. It is noteworthy that Alexander's solution contains acid fuchsin and green malachite, which react with the protoplasm and the pollen tube cell wall, which is composed of cellulose. Here are some references about the efficiency of using Alexander’s solution:

 Alexander L (2020). Front. Plant Sci. 11:100. doi: 10.3389/fpls.2020.00100

 Frescura VD et al. (2012). Biocell. 36(3):143-145. PMID: 23682430. 

 Souza EH et al. (2015). Euphytica 204: 13-28. doi.org/10.1007/s10681-014-1273-3

 Oliveira Souza S et al (2020). Microscopy Research and Technique, 84(3): 441-459. https://doi.org/10.1002/jemt.23601

 Dafni A (1992). Pollination ecology. Oxford University Press.

Line 212: Was there any particular reason for this modification in percentages or addition of PVP? It may be mentioned here. 

Response: Sorry, there was a mistake here. We did not change the CTAB protocol. We removed that information from the manuscript. 

Line 245: while there may be reproductive barriers among the accessions crossed in the experiment, the outcome would have been different in had the pollen viability and stigma receptivity test been done in accordance with the standard methods. Wrong or no assessment of viability has been one of the reasons of wrong judgements/conclusion in the past (see reference below) American Journal of Botany Vol. 82, No. 9 (Sep., 1995), pp. 1186-1197 (12 pages)

Response: 

Please note that we analyzed the stigma receptivity using standard in vivo testing considering pre-anthesis, anthesis, and post-anthesis period. In addition, a previous study demonstrated that stigmas remain receptive for up to three days after anthesis in cassava (doi: 10.1080/19420889.2019.1631110). We made some changes in the “Material and methods” section to clarify this statement. 

In addition, we used aniline blue staining to verify the presence of reproductive barriers, while to verify the pollen grain germination and the development of pollen tubes along the pistil, we used fluorescence microscopy with an ultraviolet filter, as follows: “The material was stained overnight with aniline blue solution (0.01%) in tribasic phosphate buffer. To verify the germination of pollen grains in the stigma and the development of pollen tubes along the pistil, fluorescence microscopy with ultraviolet filtering was used (25,26,18). The slides were analyzed and photographed under a BX51 fluorescence microscope (Olympus Latin America Inc.).”

We also performed a complementary analysis to evaluate the pollen grains’ viability by genotype using the histochemistry with Alexander's solution, which is comparable with other methods with results quite similar to in vitro and in vivo germination tests (https://doi.org/10.1002/jemt.23601).

Line 279: Please see my comments above.

Response: Ok!

Line 339:...further growth of the tube in the stylar region.

Response: Ok, thanks for this suggestion. 

Line 341-342: Not clear. Does this figure refer to 0.38% of the crosses? if so then it is obvious that pollen tubes have failed to reach the ovules.

Response: We changed the sentence to: “The cessation of pollen tube growth in the stylet was observed in only 0.38% of crosses (Fig 4G), and no pollen tubes were observed in the ovary region or penetrating the micropyle (Fig 4K–L).”.

Line 354: In manual deposition of pollen the PGG values are likely to be high. How the adherence was tested. Did the authors wash the stigma after certain duration of pollen deposition? Among incompatible crosses most of the pollen will not adhere and would be removed in washing.

Response: The adherence of pollen grains was evaluated by fluorescence microscopy, by counting the number of pollen grains adhered to the stigma surface considering the scale: few (<10 pollen grains), medium (10 to 25 pollen grains) and many grains (> 25 pollen grains). Pistil washing was not performed, and the methodology was used for all analyzed genotypes.

Line 417-423: Did the cultivars varied in terms of outcomes of unilateral vs bilateral crossings?

Response: The progenitors were crossed as male and female whenever possible. However, most crosses did not have a reciprocal, due to lack of synchronism of female or male flowering. Due to the high imbalance, we only conducted unilateral crossing.

Line 432: In my opinion, the data generated form breeding experiments is based on some flawed methods on pollen viability and not selecting a suitable stigmatic stage for crossing experiments.

Response: We adopted methods widely used in floral biology studies and compatibility in breeding programs. It is worth mentioning that the receptivity of the stigma was associated with a strong enzymatic reaction from anthesis to post-anthesis, which was proven by the stigma receptivity test (added in this new version of the manuscript). We certified that the crossings carried out in this work were performed with the highest pollen viability and stigma receptivity.

Line 470: Ovule or seed abortion should be categorised as the consequence of hybrid sterility (post-zygotic) and crosses involving abortion of flowers due to pollen tube inhibition before syngamy as pre-fertilization barriers.

Response: We changed the text to: “Since the ovule or seed abortion is a consequence of hybrid sterility (post-zygotic) and crosses involving abortion of flowers due to pollen tube inhibition before fertilization as pre-fertilization barriers, we cannot accurately conclude whether the reproductive barriers observed were pre- or post-zygotic.”

Line 516: Stylar inhibition of pollen tubes would mean a gametophytic control rather than sporophytic one.

Response: We totally agree with the comment, since this event is commonly related to gametophytic control. However, the ratio of 0.38% is too low to be meaningful. Probably this event could be related to an unknown factor. That is why we worked with the evidence of partial self-incompatibility. Additionally, more studies involving locus SI should be conducted to evaluate incompatibility in cassava. 

Line 519: Poor callose plug deposition due to incompatible reaction is consequence rather than a cause of slow tube growth rate manifested, which in turn is resulted from incompatible reaction in the pistil. 

Response: Thanks for the comments. We changed the sentence to “In tomato, poor deposition of callose plates is a consequence of slower pollen tube growth in plants with reduced pollen receptor kinase (LePRK2) expression (46).”

Lines 539-541: Slower pollen tube growth rate in the pistil in such crosses may be due to several reasons including pollen vigour, incompatibility reaction. pollen grains with moderate viability also grow slowly and then fail to reach the pistil while the vigorous ones grow faster.

Response: Thanks for the suggestion. It was added.

Line 555: Do you mean to say hybrid sterility or seed viability?

Response: We meant to say seed viability.

Lines 557-564: Why this result has been discussed in the context of receptivity so late in the MS? It should be mentioned right from the Methods onwards that in vivo test was done to test the receptivity. 

Response: Thanks for this suggestion. We have reorganized the presentation of this information in both “Results” and “Discussion” sections.

Line 572: It means that the stigma has cuticle pellicle layer that is ruptured only post anthetically, which explains the higher germination rate of pollen at those stages. However, this needs to be verified histochemically. 

Response: We rewrote the sentence to “Despite applying a standardized procedure for deposition of pollen grains during controlled pollinations, there was wide variation in the number of pollen grains adhered to the stigma surface among male parents. This can be explained because the stigmatic cuticle ruptures only after anthesis, which explains the higher germination rate of pollen at that stage. However, this needs to be verified by histochemical studies. Another hypothesis is…”

Line 575: ...and glycoproteins such GRP17 as in Arabidopsis thaliana and arabinogalactans.

Response: We included the suggestion in the manuscript.

Line 601: Is there any information on the allelic diversity at the SI locus in Cassava. If there are many alleles, then there could be dominance relationship in play effecting diverse outcomes on fertilization success and seed set.

Response: So far there is no information about allelic diversity at the SI locus in cassava. This is an area that deserves more in-depth studies.

Reviewer #2: Comments: PONE-D-21-16565.pdf

The manuscript describes a work which is not very commonly pursued. The experiments were designed meticulously to test the hypotheses. However, the MS needs to be improved significantly for its English language and expressions so that it is comprehensible to a reader without any ambiguity and without any need for speculation by the reader. e.g. Line 435- "Combinations between parents of the same group produced the highest numbers of pollen grains adhering to the surface of the stigma"- Here most probably the authors want to state that when crosses were made between parents of the same group, a higher number of pollen grains remained adhered to the surface of stigma. Therefore, the English of the MS needs to be improved and made more easily comprehensible. Further, following are my comments which the authors may clarify and incorporate in the MS.

Response: Thanks to the reviewers for their thoughtful comments. We submitted the manuscript to another round of proof-reading and editing to improve clarity. 

1. Line 242- "Six progenies from crosses between 10 different parents had a 100% abortion rate". Do you mean six cross combinations that involved 10 genotypes as 

Response: We meant: "Six progenies from crosses between 10 different parents had a 100% abortion rate. “. Thanks for the clarification. 

2. Line 295- How do you distinguish between pollen viability and pollen germination?

Response: We evaluated the pollen viability using histochemical pollen grain analysis. The quality of the pollen grains is assessed based on their viability and vigor. Pollen vigor refers to the speed of pollen grain germination and the rate of pollen tube growth. In addition, pollen germination is performed in in vitro tests to determine the percentage of pollen grain germination and can also be used for assessing pollen viability and vigor by monitoring the rate of germination over a period of time or the length of pollen tubes. In general, there is a linear relationship between pollen viability and germination ability in many species.

3. Line 160- "After selecting these genotypes, another 135 random 161 cross combinations were performed using 1,037 female flowers". Do you mean controlled crosses for 135 parental combinations were made? In that case, you may like to remove the word, "random".

Response: Ok, the word “random” was removed.

4. Line 182- "4) growth of the pollen tube close to the ovary". Do you mean ovary or ovule in this sentence?

Response: We meant ovary.

5. What was the depth threshold for calling the GBS-SNPs during the TASSEL pipeline?

Response: We added the information in “Material and methods”.

6. Did the authors find any set of male parents with which the seed set was significantly higher than rest of the males. Did these male parents belonged to the same group based on SNP data?

Response: Some nested effect of male x female showed higher seed setting, but unfortunately there were no relationships between the clustering based on SNP data.

7. Line 489- Is the data on relative humidity available for Expt 1? Was there any correlation of seed set rate and level of humidity?

Response: Yes, there are data of relative humidity available for Experiment 1: “Possible relationships of weather variables with abortion rate (〖AR〗_w) and seed setting rate (〖SS〗_w) were investigated using cubic regression based on average climatic data: temperature (minimum, average, and maximum); relative humidity (%); and accumulated rainfall for each 10-day intervals.” 

Relative humidity showed no association with the abortion or seed setting rate. Please see the sentence: “Relative humidity had no association with the abortion or seed setting rate”.

---

## [Decision Letter · Decision Letter 1]

27 Oct 2021

PONE-D-21-16565R1Reproductive barriers in cassava: Factors and implications for genetic improvementPLOS ONE

Dear Dr. de Oliveira,

Thank you for submitting your manuscript to PLOS ONE. After careful consideration, we feel that it has merit but does not fully meet PLOS ONE’s publication criteria as it currently stands. Therefore, we invite you to submit a revised version of the manuscript that addresses the points raised during the review process.

We look forward to receiving your revised manuscript.

Kind regards,

Shailendra Goel, Ph.D.

Academic Editor

PLOS ONE

Journal Requirements:

Additional Editor Comments:

The reviewers have expressed their satisfaction at the way manuscript has been revised, although Reviewer 1 has raised a small point, please address it.

Comment from Reviewer 1:

Although I am not convinced that how a non-vital stain can be used to ascertain pollen viability. in context of the large amount of data generated based on crossing, I suggest that the authors should at least clearly mention that for performing each cross, fresh pollen were taken. Fresh pollen is likely to have viability closer to fertility of pollen.

Reviewers' comments:

Reviewer's Responses to Questions

**Comments to the Author**

1. If the authors have adequately addressed your comments raised in a previous round of review and you feel that this manuscript is now acceptable for publication, you may indicate that here to bypass the “Comments to the Author” section, enter your conflict of interest statement in the “Confidential to Editor” section, and submit your "Accept" recommendation.

Reviewer #1: All comments have been addressed

Reviewer #2: All comments have been addressed

2. Is the manuscript technically sound, and do the data support the conclusions?

Reviewer #1: Partly

Reviewer #2: Yes

3. Has the statistical analysis been performed appropriately and rigorously? 

Reviewer #1: Yes

Reviewer #2: Yes

4. Have the authors made all data underlying the findings in their manuscript fully available?

Reviewer #1: Yes

Reviewer #2: Yes

5. Is the manuscript presented in an intelligible fashion and written in standard English?

Reviewer #1: Yes

Reviewer #2: Yes

6. Review Comments to the Author

Reviewer #1: Although I am not convinced that how a non-vital stain can be used to ascertain pollen viability. in context of the large amount of data generated based on crossing, I suggest that the authors should at least clearly mention that for performing each cross, fresh pollen were taken. Fresh pollen is likely to have viability closer to fertility of pollen.

Reviewer #2: (No Response)

7. PLOS authors have the option to publish the peer review history of their article (what does this mean?). If published, this will include your full peer review and any attached files.

Reviewer #1: No

Reviewer #2: **Yes: **Shashi Bhushan Tripathi

---

## [Author Response · Author response to Decision Letter 1]

8 Nov 2021

Reviewer #1: Additional Editor Comments: PONE-D-21-16565.pdf

The reviewers have expressed their satisfaction at the way manuscript has been revised, although Reviewer 1 has raised a small point, please address it.

Comment from Reviewer 1: Although I am not convinced that how a non-vital stain can be used to ascertain pollen viability. in context of the large amount of data generated based on crossing, I suggest that the authors should at least clearly mention that for performing each cross, fresh pollen were taken. Fresh pollen is likely to have viability closer to fertility of pollen.

Response: Thanks for your comment. Through “Material and methods” we emphasize the use of fresh pollen grains in the pollinations.

For example, in line 116-117: “Male flowers were collected during the morning of the anthesis day (7–9 am), while the crosses were performed between 9 am and 4 pm by distributing the fresh pollen grains on the stigmas”.

---

## [Editor Report · Decision Letter 2]

15 Nov 2021

Reproductive barriers in cassava: Factors and implications for genetic improvement

PONE-D-21-16565R2

Dear Dr. de Oliveira,

We’re pleased to inform you that your manuscript has been judged scientifically suitable for publication and will be formally accepted for publication once it meets all outstanding technical requirements.

Kind regards,

Shailendra Goel, Ph.D.

Academic Editor

PLOS ONE
---

## [Editor Report · Acceptance letter]

17 Nov 2021

PONE-D-21-16565R2 

Reproductive barriers in cassava: Factors and implications for genetic improvement 

Dear Dr. de Oliveira:

I'm pleased to inform you that your manuscript has been deemed suitable for publication in PLOS ONE. Congratulations! Your manuscript is now with our production department. 

Kind regards, 

on behalf of

Dr. Shailendra Goel 

Academic Editor

PLOS ONE